# Optimal transport analysis reveals trajectories in steady-state systems

**Stephen Zhang**[ID], **Anton Afanassiev**[ID], **Laura Greenstreet**, **Tetsuya Matsumoto**, **Geoffrey Schiebinger**[ID] *

Department of Mathematics, University of British Columbia, Vancouver, Canada

* geoff@math.ubc.ca

**Data Availability Statement:** The Arabidopsis thaliana dataset published by Shahan and Hsu et al. is available at https://github.com/Hsu-Che-Wei/ COPILOT. An implementation of the StationaryOT

## Abstract

Understanding how cells change their identity and behaviour in living systems is an important question in many fields of biology. The problem of inferring cell trajectories from single-cell measurements has been a major topic in the single-cell analysis community, with different methods developed for equilibrium and non-equilibrium systems (e.g. haemato-poeisis vs. embryonic development). We show that optimal transport analysis, a technique originally designed for analysing time-courses, may also be applied to infer cellular trajectories from a single snapshot of a population in equilibrium. Therefore, optimal transport provides a unified approach to inferring trajectories that is applicable to both stationary and non-stationary systems. Our method, StationaryOT, is mathematically motivated in a natural way from the hypothesis of a Waddington's epigenetic landscape. We implement Stationar-yOT as a software package and demonstrate its efficacy in applications to simulated data as well as single-cell data from *Arabidopsis thaliana* root development.

## Author summary

Many important biological phenomena involve populations of cells that undergo changes in behaviour over time to achieve a desired state or function. Modern experimental technologies are able to measure aspects of cell state but cannot observe a cell at more than a single instant in time, since the cell is necessarily destroyed in the measurement process. Therefore, the relationship between the present and future states of a cell, which we call its *trajectory*, must be inferred from observable data. Since biological processes are naturally noisy, we model cells as evolving following a stochastic dynamical system with growth. We show that for datasets drawn from a population of cells in equilibrium and when estimates of cell growth rates are available, cellular trajectories can be estimated by solving an optimal transport problem. We validate our method using simulated data and demonstrate an application to transcriptomic data from *Arabidopsis thaliana* root development.

This is a *PLOS Computational Biology* Methods paper.

computational method is available as an open-source software package at https://github.com/zsteve/StationaryOT.

**Funding:** This work was supported in part by a UBC Affiliated Fellowship to S.Z. (https://www.grad.ubc.ca), a Career Award at the Scientific Interface from the Burroughs Wellcome Fund (https://www.bwfund.org), an NFRF Exploration Grant (https://www.sshrc-crsh.gc.ca), and a NSERC Discovery Grant (https://www.nserc-crsng.gc.ca) to G.S. The funders had no role in study design, data collection and analysis, decision to publish, or preparation of the manuscript.

**Competing interests:** The authors have declared that no competing interests exist.

## Introduction

Biological processes at the cellular level are driven by stochastic dynamics—cellular populations evolve through time, driven by regulation at the cellular and tissue level and intrinsic noise arising from thermal fluctuations. In the context of developmental biology, these processes have been classically described by Waddington's metaphor of an epigenetic landscape [1], in which differentiating cells can be thought of as evolving from regions of high differentiation potential into valleys corresponding to differentiated cell types. In the last decade, this metaphor has evolved to be much more quantitative [2, 3]. Modern high-throughput assays such as single-cell RNA sequencing (scRNA-seq) [4, 5], scATAC-seq [6] and CyTOF [7] now allow the molecular states of thousands of single cells to be profiled in a single experiment. With the ability to make these precision measurements of cell state, new challenges emerge in analysing these new types of high-dimensional data.

Single-cell measurements are destructive in nature, so the state of any individual cell cannot be observed at more than one instant. Therefore, information about the trajectories taken by cells over time is lost and must instead be inferred from data. A large collection of trajectory inference methods have been developed in recent years [2] to address this issue. These methods broadly fall into two classes [8]: (1) methods that deal with a single stationary snapshot observed from a cellular population at equilibrium [9–11], and (2) methods that deal with a time series of snapshots from an evolving population [3, 8, 12].

Time-series experiments are a natural approach for observing biological systems where cellular populations undergo dramatic, synchronous changes, such as in embryogenesis or stem-cell reprogramming [3, 13–16]. Trajectory inference methods for time series data primarily seek to infer cellular transition events from snapshots of one timepoint to the next. On the other hand, development occurs continuously and asynchronously in many biological systems such as haematopoiesis and spermatogenesis. These systems maintain a stationary population profile across various cell types and can be thought of as being in dynamic equilibrium (i.e. steady state). Snapshots therefore capture cells from across the full progression of cell states from undifferentiated to fully differentiated cells. Trajectory inference for snapshots sampled from these steady-state systems seek to (a) infer the progression of cells in "developmental time" (commonly referred to as pseudotime) [10, 17], and (b) uncover bifurcation events or "cellular decisions" occurring in the differentiation process [9, 18].

In this paper we show that optimal transport analysis, a technique originally applied to analyse time-courses [3], may also be applied to infer cellular trajectories from a single snapshot of a population in equilibrium. Therefore, optimal transport (OT) provides a unified approach to inferring trajectories, applicable to both stationary and non-stationary systems. Our approach is theoretically justified when the trajectories are driven by a potential landscape, as in [18]. Our method has the potential for extensions to incorporate additional information such as estimates of the vector field obtained from RNA velocity methods [19, 20] or metabolic mRNA labelling [21]. When such information is available, we can recover certain aspects of non-conservative dynamics such as oscillations. Beyond transcriptomics, our method can be applied to measurements where velocity information cannot be provided, such as scATAC-seq [6] and CyTOF [7]. Finally, the output of our method could be provided to downstream methods which aim to extract high-level information from the learned single-cell transition probabilities. Two methods [21, 22] are especially applicable: CellRank [22] leverages the theory of Markov chains to find groups of cell states and uncover lineage driver genes, while Dynamo [21] can construct a continuous vector field that is amenable to dynamical systems analysis.

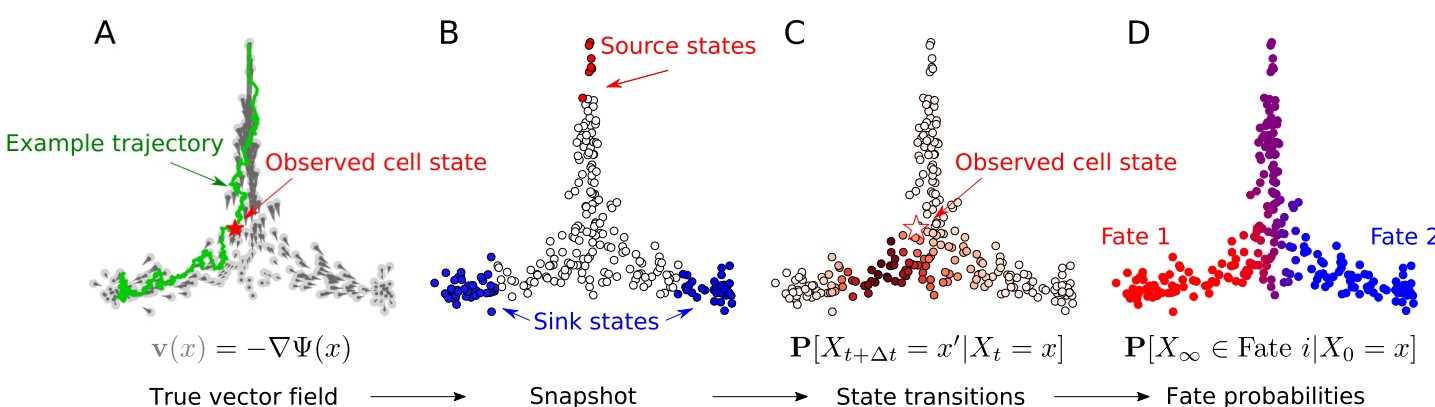

**Fig 1. Conceptual illustration of inference problem.** (a) Vector field (grey), sample trajectory (green) and observed cell state (red) drawn from ground truth process. (b) Sampled snapshot with labelled source (red) and sink (blue) states. (c) Inferred state transition probabilities. (d) Fate probabilities calculated from Markov chain.

## Modelling assumptions

**Development as drift-diffusion with birth-death.** We model cells as points in a space $\mathcal{X}$, which we take to be a representation of the space of possible cellular molecular states (for instance, in the case of scRNA-seq data, $\mathcal{X}$ represents the space of gene expression profiles). Typically, we will take $\mathcal{X} \subset \mathbb{R}^d$ to be the ambient state space. We regard cells as evolving following a drift-diffusion process [18, 23, 24] described by the stochastic differential equation (SDE)

$$\mathrm{d}X_t = \boldsymbol{v}(X_t)\mathrm{d}t + \sigma\,\mathrm{d}B_t. \tag{1}$$

where $X_t \in \mathcal{X}$ is the state of a cell at time $t$, $\boldsymbol{v}$ is a vector field, the diffusivity $\sigma^2$ captures the noise level and $\mathrm{d}B_t$ denotes the increments of a Brownian motion. Over time a cell traces out a path, or *trajectory*, $X_t$ in the state space. We illustrate these concepts in Fig 1A. In addition, cells are subject to division and death events at exponential rates $\beta(x)$ and $\delta(x)$ respectively, which may vary in spatial location in $\mathcal{X}$. That is, in an infinitesimal time interval $\mathrm{d}t$, a cell $X_t$ may divide with probability $\beta(X_t)\,\mathrm{d}t$ or die with probability $\delta(X_t)\,\mathrm{d}t$. The underlying assumption in our framework is that the evolution of cell states can be well-described by a Markov process. While in reality this property may not be truly satisfied, we note that many other methods [18, 20–22] also make this assumption for the sake of analytical tractability.

**Population-level model.** At the population level, the drift-diffusion process with birth and death can be described by a *population balance* partial differential equation (PDE) [18, 25]

$$\partial_t\rho(x,t) = -\nabla \cdot (\boldsymbol{v}(x)\rho(x,t)) + \frac{\sigma^2}{2}\nabla^2\rho(x,t) + R(x)\rho(x,t), \tag{2}$$

where $\rho(x, t)$ is a continuous population density, and $R$ is a spatially varying flux rate defined as $R(x) = \beta(x) - \delta(x)$ that captures creation and destruction of cells due to birth and death, as well as entry and exit from the system. For $R(x) > 0$ we refer to $x$ as a *source* state, and similarly for $R(x) < 0$ we refer to $x$ as a *sink* state.

**Observation model.** As we discussed earlier in this introduction, many biological processes exist approximately in an equilibrium or steady state. In this setting, a snapshot at a single instant in time will capture all stages of cellular development in the system [2, 18], and relative proportions of various cell types remain unchanged over time. Mathematically at the population level, this assumption amounts to demanding that $\partial_t\rho(x, t) = 0$ in Eq (2), that is,

the population level cell density does not change. We will write $\rho_{eq}(x)$ for this steady-state solution. Experimental observation of such a system is therefore equivalent to sampling a collection of $N$ cellular states from the population, i.e.

$$X_1, \ldots, X_N \sim \rho_{eq}.$$

We may describe this finite sample as an empirical distribution $\hat{\rho}_{eq}$ supported on the discrete space $\bar{\mathcal{X}} = \{X_i\}_{i=1}^N$,

$$\hat{\rho}_{eq} \quad = \frac{1}{N}\sum_{i=1}^N \delta_{X_i}. \tag{3}$$

Fig 1B shows an example of such a sample dataset drawn from the equilibrium distribution, where we have identified also source and sink states.

### Inference goal

**Laws on paths.**   The process described by (1) and (2) is a superposition of a birth-death process and a drift-diffusion process. The drift-diffusion component of this process, described by Eq (1), governs the evolution of individual cell states. Therefore, we seek to learn something about the drift-diffusion dynamics from observed snapshot data. In the framework of SDEs, Eq (1) (equipped also with an initial condition) induces a probability distribution over the space of possible cell trajectories. As we also argue in [23], this *law on paths* is the natural object we seek to estimate since it directly encodes the trajectories that cells may follow. In practice, since the process is time-homogeneous, it is characterised by its time-$\Delta t$ transition densities $\mathbf{P}[X_{\Delta t} \in \cdot | X_0 = x]$. We illustrate in Fig 1C the concept of a transition density in the discrete setting of Eq (3). As we will find, an approximation of this transition density for small enough $\Delta t$ can be obtained as the solution to a strictly convex minimisation problem. Conveniently, we do not have issues of multiple local minima which may be the case if we attempt to recover the drift field $\mathbf{v}$ or potential landscape $\Psi$ directly, such as in [26, 27].

**Identifiability.**   For the sake of making inferences about the law on paths induced by Eq (1), we must necessarily have estimates of the flux rate $R(x)$ and the noise level $\sigma^2$. As discussed at length by Weinreb et al. [18], when only a single snapshot (i.e. $\hat{\rho}_{eq}$) is available, in general more than one drift field $\mathbf{v}$ can give rise to the same steady-state density profile $\rho_{eq}$. To ensure uniqueness of the solution, we must restrict to the case where the drift is given by the gradient of a scalar potential [18, 23], i.e. $\mathbf{v} = -\nabla\Psi$. We note that $\Psi$ can be thought of a kind of Waddington's epigenetic landscape [1]. For completeness, we provide an example illustrating the issue of identifiability in S1 Appendix.

### Related work

The SDE framework of Eq (1) is a classical choice for modelling cell state dynamics [24, 28]. For a system at steady state with drift $\mathbf{v} = -\nabla\Psi$, solution of the corresponding Fokker-Planck equation yields a well-known relationship between the steady-state population density $\rho_{eq}$ and potential function $\Psi$:

$$0 \quad = \nabla \cdot (\rho_{eq}\nabla\Psi) + \frac{\sigma^2}{2}\nabla^2\rho_{eq}$$

$$\Rightarrow \Psi(x) \quad = -\frac{\sigma^2}{2}\log\rho_{eq}.$$

In practice, a potential landscape can be reconstructed using this relationship if the steady-

state density $\rho_{eq}$ can be estimated from samples, typically using techniques such as kernel density estimation [26, 28], although this can be difficult in high dimensions. This approach, however, ignores cell growth and death because the PDE above lacks a source term on the left-hand-side. This observation motivates the addition of the flux rates $R(x)$ to form a more general model as we have done in Eq (2). Once flux is added to the model, a different steady-state solution will be achieved.

In an alternative methodological direction, a significant amount of work has been devoted towards the problem of recovering the topology of the dynamics [11, 29, 30]. This can be thought of as a coarse-grained approach that is concerned with uncovering features such as bifurcations. While extracting the topology is certainly powerful and highly interpretable, there is an inherent loss in resolution, since these methods do not truly estimate dynamics at the level of single cells. For this reason, we consider in this paper the Fokker-Planck framework at the level of single cells.

Weinreb et al. [18] previously investigated the inference problem for this model and discussed at length the need for a gradient system for identifiability. In addition, the authors presented population balance analysis (PBA), a methodological framework for estimating the potential $\Psi$ based on spectral graph theory. Although our approach and that of [18] share a problem formulation and may indeed perform similarly, we note that the theoretical foundations of the two approaches are fundamentally distinct—our method is based on solving a convex optimisation problem for the transition probabilities, whilst PBA solves a system of linear equations for the potential. As an optimisation-based method, optimal transport also allows for incorporation of additional information such as velocity estimates.

Optimal transportation (OT) theory is a mathematical area of study concerned with optimally coupling probability distributions [31] which has recently found diverse applications in statistics, machine learning and computational geometry. Optimal transport has been applied to the problem of tracking particle ensembles [32, 33], and to single-cell trajectory inference in the setting of time-series population snapshots in [3, 26]. Subsequent work has extended both methodology and theory in this direction, e.g. [23, 27, 34–36]. However, these works focus on the setting where multiple snapshots are available over a series of time-points. We show in this work that optimal transport can be applied in a natural way to the case of a single stationary snapshot, further establishing optimal transport as a widely applicable and robust framework for single-cell trajectory inference.

## Results

### Overview of results

To motivate the mathematical framework for our method, we will consider first the population-level setting of infinitely many cells. We then reduce this to the discrete setting where we deal with finite samples drawn from the steady state population. We name our method StationaryOT and implement it as a software package. Next, we apply the method to simulated datasets sampled from drift-diffusion processes in the setting of both potential-driven and non-conservative vector fields. Finally, we demonstrate an application to a stationary snapshot scRNA-seq dataset in *Arabidopsis thaliana* root tip development and discuss approaches for applying StationaryOT to very large datasets by utilising GPU acceleration, showing that our method can scale to $1.1 \times 10^5$ cells with runtimes of $\sim 1$ hour.

Throughout this paper, we consider using either entropy-regularised or quadratically-regularised optimal transport for the main step of the StationaryOT method. Although entropy-regularised optimal transport is the one that arises naturally from the theoretical motivation, we demonstrate in practice that using the quadratic regularisation generally leads to results

that are more robust and interpretable, as well as being more computationally favourable thanks to sparsity of the recovered transition laws, without substantial sacrifices to accuracy.

## Methodology: Population level

At the steady state of the process described by Eq (2), the population density profile is constant, i.e. $\rho(\cdot, t) = \rho_{\mathrm{eq}}$. However, at the microscopic level, individual cells $X_t$ continue to undergo drift-diffusion according to Eq (1), as well as birth-death. Thus, observation of population profiles in the stationary setting do not contain information about the dynamics of individual particles, unlike the non-stationary setting of time series measurements [3, 23].

Suppose that we are able to observe a cell $X_t$ from the stationary population at time $t$, and again at time $t + \Delta t$ (conditioned on not dividing, dying or exiting from the system in that time interval). Then the joint distribution $(X_t, X_{t+\Delta t})$ would capture information about all possible transitions in cell state over a time interval $\Delta t$. The system is at a steady state and the dynamics are Markov, so knowledge of the time-$\Delta t$ evolution of the system captures the full law on paths that results from the drift-diffusion component, at least at times $\{k\Delta t, k = 0, 1, \ldots\}$ by simply composing Markov transitions.

Since we may access densities but not track individual particles, we cannot measure the joint distribution $(X_t, X_{t+\Delta t})$ directly. Therefore, we seek to infer it from observation of a single snapshot $\rho_{\mathrm{eq}}$ and information about the birth-death rates as well as noise level. In the underlying process both birth-death and drift-diffusion take place simultaneously, leading to complications in directly reasoning with probability laws. In order to simplify this, we approximate the evolution of the process by introducing an artificial separation of the effects of growth and transport, inspired by operator splitting methods from numerical analysis [37]. That is, we split the linear equation Eq (2) into equations corresponding to growth and transport in the densities $\rho_G$ and $\rho_T$ respectively:

$$\frac{\partial \rho_G}{\partial t} = R(x)\rho_G(x, t), \tag{4}$$

$$\frac{\partial \rho_T}{\partial t} = -\nabla \cdot (\boldsymbol{v}(x)\rho_T(x, t)) + \frac{\sigma^2}{2}\nabla^2 \rho_T(x, t) \tag{5}$$

$$\text{where } \rho_G(\cdot, 0) = \rho_{\mathrm{eq}}(\cdot) \text{ and } \rho_T(\cdot, 0) = \rho_G(\cdot, \Delta t). \tag{6}$$

Then $\rho_T(\cdot, \Delta t)$ is a splitting approximation of the true steady-state solution $\rho(\cdot, \Delta t) = \rho_{\mathrm{eq}}(\cdot)$ of Eq (2), and the two coincide in the limit $\Delta t \to 0$ with pointwise approximation error of order $\mathcal{O}(\Delta t^2)$ [37, Section 1.3], i.e.

$$\rho_T(x, \Delta t) = \rho(x, \Delta t) + \mathcal{O}(\Delta t^2), \ x \in \mathcal{X}. \tag{7}$$

We provide a conceptual illustration of this scheme in Fig 2. The solution of Eq (4), corresponding to the growth step, can be determined to be exactly

$$\rho_G(x, \Delta t) = \rho(x, 0)e^{\Delta t R(x)} = \rho(x, 0)g(x)^{\Delta t},$$

where we have taken $g(x) = e^{R(x)}$.

It therefore remains for us to examine the effects due to transport as described by Eq (5). Since the overall system is assumed to be at steady state, composing the effects of growth and transport should yield the initial density $\rho(\cdot, 0) = \rho_{\mathrm{eq}}(\cdot)$ up to the $\mathcal{O}(\Delta t^2)$ error introduced by the splitting approximation. For brevity, let us denote $\mu_0 = \rho_T(\cdot, 0)$ and $\mu_1 = \rho_T(\cdot, \Delta t)$ to be the distributions before and after transport. Under the splitting approximation, our problem of

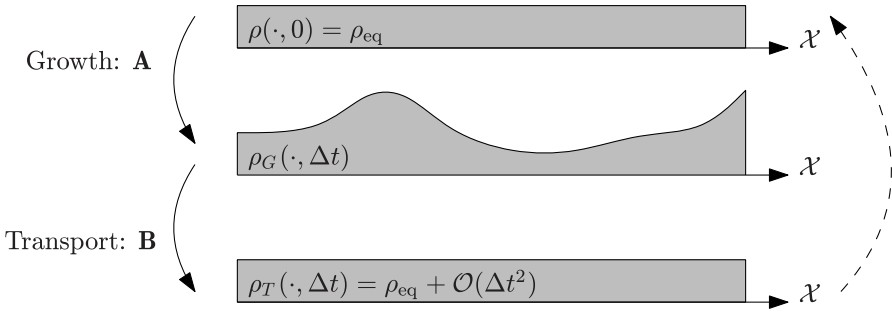

**A.** $\partial_t \rho_G = R(x)\rho_G(x,t), \; \rho_G(\cdot,0) = \rho(\cdot,0)$
**B.** $\partial_t \rho_T = -\nabla \cdot (\mathbf{v}(x)\rho_T(x,t)) + \frac{1}{2}\sigma^2 \nabla^2 \rho_T(x,t), \; \rho_T(\cdot,0) = \rho_G(\cdot,\Delta t)$

**Fig 2. Illustration of the splitting scheme for decomposing Eq (2) into growth (Eq (4)) and transport (Eq (5)).** Composing the effects of growth and transport must maintain the steady-state profile $\rho_{\text{eq}}$. The coupling induced by transport is recovered by matching $\rho_G(\cdot,\Delta t)$ and $\rho_T(\cdot,\Delta t)$.

estimating the joint law $(X_t, X_{t+\Delta t})$ conditional on no birth or death events amounts to finding an appropriate coupling $\gamma_{\Delta t}$ of $(\mu_0, \mu_1)$, i.e. a joint distribution $\gamma_{\Delta t}$ on $\mathcal{X}^2$ whose marginals agree with $\mu_0$ and $\mu_1$. We note that the contribution of transport to the dynamics involves drift and diffusion, which scale as $\mathcal{O}(\Delta t)$ and $\mathcal{O}(\sqrt{\Delta t})$ respectively. Since the error incurred by separating the effects of growth and transport scales as $\mathcal{O}(\Delta t^2)$, the scheme is asymptotically consistent.

## Inference by optimal transport

By the previous construction, we seek to couple the distributions $(\mu_0, \mu_1)$ in a way that approximates the "true" underlying transition law $(X_t, X_{t+\Delta t})$. To be concise, we write

$$\Pi(\mu_0, \mu_1) = \{\pi \in \mathcal{M}_+(\mathcal{X} \times \mathcal{X}) : \int \pi(dx, \cdot) = \mu_1, \int \pi(\cdot, dy) = \mu_0\}$$

to denote the set of possible couplings between $\mu_0$ and $\mu_1$. For a set of prescribed marginals there are in general many valid couplings: indeed, for any $\mu_0, \mu_1$ we may always construct the independent coupling, $\mu_0 \otimes \mu_1$. Therefore, additional assumptions on the nature of the process driving the evolution from $\mu_0$ to $\mu_1$ are needed if we desire a unique "best" coupling.

From the drift-diffusion step of Eq (5), we know that the evolution from $\mu_0$ to $\mu_1$ is described by a drift-diffusion equation (with no source term). At the level of individual particles, this is equivalent to Eq (1). We note further that for $\Delta t$ small, the effect of the drift component is $\mathcal{O}(\Delta t)$ and is therefore drowned out by the effect of the diffusion component which is $\mathcal{O}(\sqrt{\Delta t})$. Thus, for small $\Delta t$, the setting which we approach is that of a diffusive evolution in time $\Delta t$ from $\mu_0$ to $\mu_1$, and the most likely coupling $\gamma_{\Delta t}$ is unique and is characterised by an entropy minimisation principle that is well known in the literature of optimal transport and large deviation theory [38]. Specifically, the optimal coupling $\gamma_{\Delta t}$ is the minimiser of the so-called *Schrödinger problem*:

$$\min_{\gamma_{\Delta t} \in \Pi(\mu_0, \mu_1)} \mathrm{H}(\gamma_{\Delta t} | K_{\sigma^2 \Delta t}). \tag{8}$$

In the above, $\mathrm{H}(\alpha|\beta) = \int d\alpha \log\left(\frac{d\alpha}{d\beta}\right)$ is the relative entropy between distributions, and $K_{\sigma^2 \Delta t}$ is

the kernel

$$K_{\sigma^2 \Delta t}(x, y) = \exp\left(-\frac{1}{2\sigma^2 \Delta t} \|x - y\|^2\right),$$

corresponding to the time-$\Delta t$ evolution of a Brownian motion in $\mathcal{X}$ with diffusivity $\sigma^2$.

The problem Eq (8) is also known in the optimal transport literature as entropy-regularised optimal transport [31], where the objective to be minimised is often written in the alternative form

$$\min_{\gamma_{\Delta t} \in \Pi(\mu_0, \mu_1)} \int C(x, y) \mathrm{d}\gamma_{\Delta t}(x, y) + \varepsilon \mathrm{H}(\gamma_{\Delta t} | \mathrm{Leb}) \tag{9}$$

where $C(x, y) = \frac{1}{2} \|x - y\|^2$ is a quadratic cost function, $\varepsilon = \sigma^2 \Delta t$ is the entropy regularisation parameter and Leb is the reference Lebesgue measure on $\mathcal{X}$. Written in this way, Eq (9) can be understood as a least action principle, where the optimal $\gamma_{\Delta t}$ is roughly the one that minimises the expected action for moving mass from $\mu_0$ to $\mu_1$, if the action is proportional to the squared distance moved. In the limiting case of vanishing noise where $\varepsilon \to 0$, the entropy-regularised optimal transport problem becomes what is known as the Monge-Kantorovich problem, or unregularised optimal transport [31].

We conclude that the coupling $\gamma_{\Delta t}$ recovered by solving the entropy minimisation problem Eq (8) is an approximation to the true evolution of Eq (1), corresponding to the drift-diffusion step Eq (5) of the splitting scheme. This connection between entropy-regularised optimal transport and SDEs dates back to the work of Schrödinger [39] (see [38] for a general survey, and see [23, Theorem 2.1] for a more detailed discussion).

## Methodology: Finite samples

**Formulation of the discrete problem.**   In practice, we have access to an empirical distribution $\hat{\rho}_{\mathrm{eq}}$ (see Eq (3)) supported on the discrete set $\bar{\mathcal{X}}$ that can be thought of as approximating the true continuous density $\rho_{\mathrm{eq}}$ discussed previously. We also assume for each observed cell $x_i$ that we have an estimate of the corresponding flux rate $\hat{R}_i = R(x_i) = \beta(x_i) - \delta(x_i)$. In a practical biological setting, cell states which are expected to divide or die should therefore have $\hat{R}_i > 0$ or $\hat{R}_i < 0$ respectively, and those states which do neither should have $\hat{R}_i = 0$. In addition to division and death, terminally differentiated cells expected to shortly exit the system may be regarded as representing sinks, and therefore assigned $\hat{R}_i < 0$. The numerical values for flux rates may be estimated from cell-cycle signatures [3] or prior biological knowledge [18].

In this discrete setting, the growth step Eq (4) is local in space and thus its analogue can be directly written for a chosen small value of $\Delta t$ to obtain $\mu_0$:

$$\mu_0(x_i) = \hat{\rho}_{\mathrm{eq}}(x_i) e^{\Delta t \hat{R}_i} = \hat{\rho}_{\mathrm{eq}}(x_i)(1 + \Delta t \hat{R}_i + \mathcal{O}(\Delta t^2)). \tag{10}$$

Next, the effect of the transport step Eq (5) is to rearrange mass via diffusion and drift so that we return to the steady state distribution $\hat{\rho}_{\mathrm{eq}}$. We cannot take $\mu_1$ to be $\hat{\rho}_{\mathrm{eq}}$ exactly, since a single step of the splitting scheme introduced in Eqs (4) and (5) is only accurate up to $\mathcal{O}(\Delta t^2)$. Therefore, a straightforward application of the growth step Eq (10) will result in a slight change in the total mass of the system. Additionally, in practice we have only estimates $\hat{R}_i$ of the true flux rates, further contributing to this effect. Consequently, we must instead re-normalise $\mu_1$ so that it has the same mass as $\mu_0$:

$$\mu_1(x_i) \propto \hat{\rho}_{\mathrm{eq}}. \tag{11}$$

With $\mu_0$ and $\mu_1$ constructed in this way, we may compute the solution to the discrete Schrödinger problem Eq (8).

**Choice of $\varepsilon$ and $\Delta t$.**   The key parameters for the scheme we describe are $\Delta t$, the time step introduced in the growth splitting, and the regularisation parameter $\varepsilon$ for entropy-regularised optimal transport. In the theoretical framework of the Schrödinger problem, these parameters have a proportional relationship $\varepsilon = \sigma^2 \Delta t$. The accuracy of the scheme should improve in the limit as $\varepsilon \to 0$, $\Delta t \to 0$ and $\varepsilon = \sigma^2 \Delta t$, since the splitting approximation becomes exact. However, in practice where we have discrete samples we find that allowing $\varepsilon$ and $\Delta t$ to deviate from this relationship often leads to better results.

In the discrete setting, a key limitation is the value of $\varepsilon$, which controls the level of diffusion in the reference process in the Schrödinger problem, and consequently influences the level of diffusion in the inferred process. In practice when we are dealing with a limited number of samples in a potentially high-dimensional space, taking $\varepsilon$ too small may lead to an ill-conditioned problem. The reason for this is that the distance between points in the set of samples $\bar{\mathcal{X}}$ may be quite large compared to $\varepsilon = \sigma^2 \Delta t$. That is, $\exp\left(\frac{-\|x-y\|^2}{2\sigma^2 \Delta t}\right)$ may be exceedingly small, resulting in a reference process that mixes extremely slowly. On the other hand, if we pick a reasonably sized $\varepsilon$, strictly adhering to the proportionality relationship may mean that the corresponding $\Delta t$ is too large for the splitting approximation to be a good one. In practice, we have often found that it is helpful to take $\Delta t$ to be slightly smaller (and consequently $\varepsilon$ slightly larger) than what is expected in theory.

## Quadratically regularised optimal transport

The entropy-regularised optimal transport problem Eq (9) is well known for its probabilistic interpretation and the existence of an efficient solution scheme by matrix scaling [31]. However, the use of entropic regularisation results in a transport plan that necessarily has a dense support [40]. Recent contributions to the optimal transport literature [40, 41] have highlighted that alternative choices of the regulariser may yield other smooth approximations of the Monge-Kantorovich problem which exhibit desirable properties. In particular, using a quadratic ($L^2$) regulariser to form the problem

$$\min_{\gamma \in \Pi(\mu_0, \mu_1)} \int C(x,y) d\gamma(x,y) + \varepsilon \|\gamma\|_2^2 \tag{12}$$

gives rise to what is known as the *quadratically* regularised optimal transport problem. As noted by [40, 41], quadratically regularised OT has the property that transport plans are generally sparse in practice (in the discrete case, transition probabilities are nonzero only on a sparse graph that spans the data), making it a favourable choice for interpretability of transport plans as well as computational efficiency. In addition, in [40] the authors remark that the quadratically regularised problem may be less prone to issues of numerical stability.

In practice, we may employ a quadratic regularisation in our scheme by substituting the solution of Eq (12) for the optimal coupling $\gamma_{\Delta t}$ instead of the entropy-regularised solution Eq (8). As we later demonstrate, we find evidence that quadratic regularisation is more robust to parameter choices and noise compared to entropy regularisation.

## Extension to non-potential vector fields

Estimation of dynamics in the case where the underlying drift $v$ does not arise from a potential gradient requires additional information to be available, such as potentially noisy or partial estimates of the velocity of cells [19–22]. Since at its core our method is based on solving a

convex optimisation problem, additional information such as velocity estimates can be incorporated into our estimation procedure in a straightforward manner by modifying the cost matrix $C$. Indeed, suppose for each observed cell $x_i$ we also have an estimate of its velocity $v_i$. In the setting of velocity estimates derived from RNA velocity, the orientation of velocity estimates is more biologically informative than the magnitude [27], and it is therefore natural to incorporate velocity information in terms of cosine similarities [20, 22]. In our case, we consider an overall cost function that is a linear combination of the standard squared Euclidean cost $C_{\mathrm{euc}}$ and a matrix of cosine similarities $C_{\mathrm{velo}}$, i.e.

$$C = \lambda_1 C_{\mathrm{euc}} + \lambda_2 C_{\mathrm{velo}},$$

where

$$(C_{\mathrm{velo}})_{ij} \quad = \frac{1}{2}\left(1 - \frac{\langle x_j - x_i, v_i \rangle}{\|x_j - x_i\|\|v_i\|}\right). \tag{13}$$

In practice, the weights $\lambda_1, \lambda_2$ would depend on the relative scales of $C_{\mathrm{euc}}$ and $C_{\mathrm{velo}}$, as well as any cost normalisation that is applied.

## Simulated data—Potential driven dynamics

**Simulation setup and parameters.** We first consider a tri-stable system (1) in $\mathcal{X} = \mathbb{R}^{10}$, with drift term $v$ taken to be the negative gradient of the potential

$$\Psi(x) = 2.5\|x - z_0\|^2\|x - z_1\|^2\|x - z_2\|^2, \tag{14}$$

with wells $\{z_0, z_1, z_2\}$ located at

$$
\begin{aligned}
z_0 &= 1.05[\cos(\pi/6), \sin(\pi/6), 0, \ldots, 0]^\top \\
z_1 &= 1.05[\cos(5\pi/6), \sin(5\pi/6), 0, \ldots, 0]^\top, \\
z_2 &= 1.05[\cos(-\pi/2), \sin(-\pi/2), 0, \ldots, 0]^\top.
\end{aligned}
$$

We illustrate this potential landscape in Fig 3A in the first two dimensions of $\mathcal{X}$. Simulated particles are initially isotropically distributed around the origin following the law

$$X_0 \sim 0.01\mathcal{N}(0, I) \tag{15}$$

at $t = 0$, where $\mathcal{N}(0, I)$ denotes the standard normal distribution in $\mathbb{R}^{10}$ with covariance $I$. Particles then evolve following drift-diffusion dynamics with $\sigma^2 = 0.5$. Whenever a particle falls in the vicinity of any of the potential wells $\{z_0, z_1, z_2\}$, it is removed with exponential rate 5. That is, in each time step $\mathrm{d}t$, a particle located in a sink region is removed with probability $5\,\mathrm{d}t$. We defined the sink region for each potential well $z_i$ to be a ball of radius $r = 0.25$ centred at $z_i$.

Exact sampling of snapshots from the steady state distribution $\rho_{\mathrm{eq}}$ of Eq (2) would require the solution of a high-dimensional PDE and is therefore computationally difficult. Instead, we obtain an approximate snapshot of the system at its steady state by simulating $N = 250$ trajectories from start to finish using the Euler-Maruyama method

$$X_{t+\tau} = X_t - \tau\nabla\Psi(X_t) + \sigma\sqrt{\tau}\mathcal{N}(0, 1). \tag{16}$$

For our simulations we employed a time step $\tau = 1 \times 10^{-3}$, and from each trajectory $\{X_t^{(i)} : 0 \le t \le T_{\mathrm{final}}^{(i)}\}, 1 \le i \le N$ we sampled a single particle state chosen at a random time chosen uniformly on $[0, T_{\mathrm{final}}^{(i)}]$ to form the snapshot data $\hat{\rho}_{\mathrm{eq}}$. This scheme was also the one used for obtaining snapshots in [18].

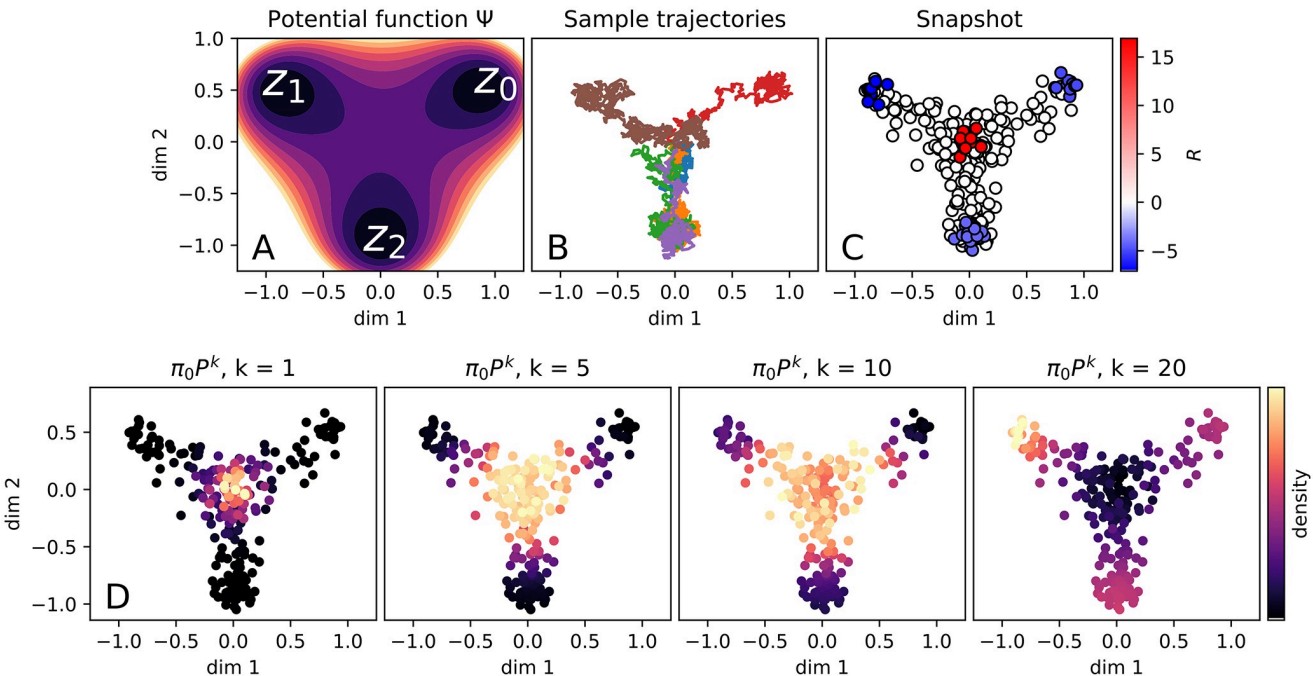

**Fig 3. Potential-driven simulation.** (a) Illustration of the potential $\Psi$ in the first two dimensions of the space $\mathcal{X}$. (b) Examples of simulated particle trajectories $X_t^{(i)}$ following the drift-diffusion process. (c) Snapshot particles $\hat{\rho}_{eq}$ shown in the first two dimensions of $\mathcal{X}$, with the value of $R$ indicated. Source and sink regions correspond to $R > 0$ and $R < 0$ respectively. (d) Evolution of the dynamics recovered by StationaryOT.

Particles $x_i$ located in the sink regions were labelled as 'sink' sites and assigned flux rates $R_i$ so that the average sink flux rate was $-5$ and total flux rate for each well $\{z_0, z_1, z_2\}$ matched the ground truth in proportion. Particles located in a ball of radius 0.25 of the origin were labelled as 'source' sites, corresponding to locations $x_i$ with $R_i > 0$. Since we deal with finite samples, we assigned $R_i$ uniformly on source sites such that the equilibrium condition $\Sigma_i R_i = 0$ was satisfied.

We display some example trajectories in Fig 3B, and illustrate the snapshot data $\hat{\rho}_{eq}$ in Fig 3C, where the values of $R_i$ at source and sink sites are shown by colour.

**Inferring dynamics using StationaryOT.**   To apply StationaryOT, we chose a time step $\Delta t = 25\tau = 2.5 \times 10^{-2}$, noting that this is small compared to the average particle lifespan of 0.934 in this simulation. We solved the StationaryOT problem using entropy-regularised optimal transport using a range of regularisation parameter values $\varepsilon$ in $10^{-2.5} - 10^1$. As we discuss in more detail later, we found that $\varepsilon = 0.026$ best matched the ground truth in terms of average fate probability correlation across the three lineages. For this choice of $\varepsilon$ we computed a forward transition matrix $P$ from the optimal transport coupling $\gamma_{\Delta t}$ by row-normalising:

$$P_{ij} = \frac{(\gamma_{\Delta t})_{ij}}{\sum_j (\gamma_{\Delta t})_{ij}}.$$

The matrix $P$ therefore describes a time-$\Delta t$ evolution of probability densities on the discrete set $\bar{\mathcal{X}}$. For an initial distribution $\pi_0$ supported on $\bar{\mathcal{X}}$, we can compute the evolution $\{\pi_0 P^k, k = 0, 1, 2, \ldots\}$ over steps of length $\Delta t$, which we take to be an estimate of the dynamics of the underlying drift-diffusion process. In Fig 3D we show the inferred process for $k = 1, 5, 10, 20$ where we have taken $\pi_0$ to be uniform on the source sites.

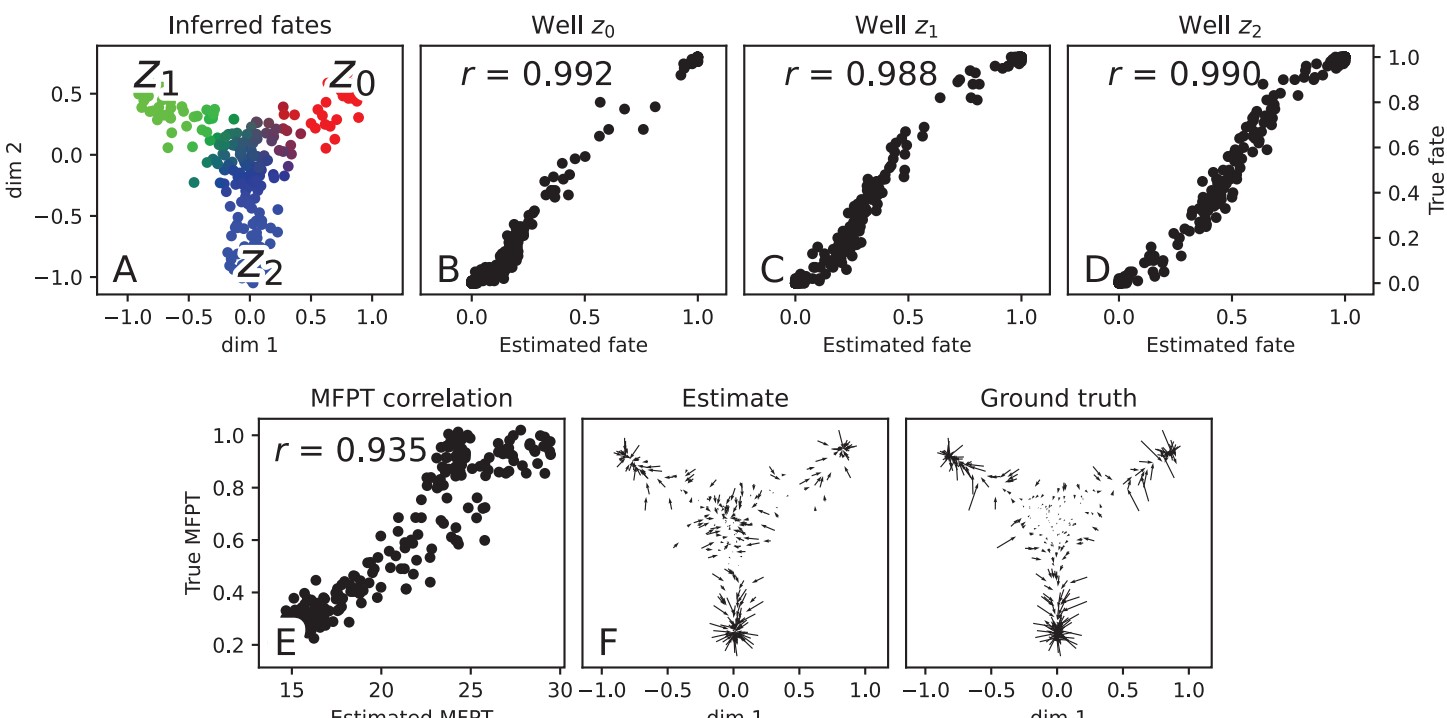

**Fig 4. Accuracy of inferred dynamics for potential-driven system.** (a) Colours representing estimated fate probabilities towards each of the wells $\{z_0, z_1, z_2\}$ are displayed on the snapshot coordinates. (b-d) Correlation with ground truth fate probabilities. (e) Comparison of estimated MFPT (in terms of Markov chain steps) to ground-truth MFPT (in continuous time units). (f) Comparison of recovered velocities to ground truth velocity.

From the transition probabilities $P_{ij}$ we may compute fate probabilities for each of the three lineages defined by the potential wells $\{z_0, z_1, z_2\}$. (These are absorption probabilities of the Markov chain $P$—see S1 Appendix for further details). We summarise these fate probabilities in Fig 4A–4D, and find that the correspondence between inferred and ground truth fate probabilities measured in terms of the Pearson correlation is high ($r \approx 0.99$). As another measure of the accuracy of the estimated dynamics, we compute the mean-first passage time (MFPT) of each sampled point $x_j$. This is the expected time at which a Markov chain initialised at a randomly chosen source location $x_i$ reaches $x_j$:

$$\text{MFPT}(x_j) = \mathbb{E}_{x_i \sim \text{sources}} \text{MFPT}(x_j | x_i),$$

where $\text{MFPT}(x_j | x_i)$ denotes the conditional MFPT for a particle starting at $x_i$ to hit state $x_j$. Comparing the MFPT estimates to the ground truth MFPT in Fig 4E, we find that the correspondence is high ($r > 0.9$).

**Reconstructing the drift field $v$.** Since the transition probabilities encode the displacement law of the underlying process over a time interval $\Delta t$, we can also recover an estimate $\hat{v}$ of the velocity field $v$ by computing the expected time-$\Delta t$ displacement of each cell:

$$\hat{v}(x_i) = \frac{\mathbb{E}_P(X_{\Delta t} - X_0 | X_0 = x_i)}{\Delta t}.$$

In Fig 4F we show the estimated velocity field $\hat{v}$ alongside the ground truth $v$, and we measure

the error by computing the mean cosine error between vector fields:

$$\frac{1}{N}\sum_{i=1}^{N}(1 - \cos\angle(\boldsymbol{v}(x_i), \hat{\boldsymbol{v}}(x_i))) \approx 0.024.$$

We observe that the estimated field $\hat{\boldsymbol{v}}$ resembles the ground truth quite well near the potential wells where particles are subjected to a relatively strong drift, but struggles near the origin where the true velocity field has a small magnitude. Overall however, the cosine error is close to zero, indicating that our recovered velocity field matches the ground truth field well.

**Effect of the choice of regularisation parameter $\varepsilon$ and flux rates $R$.**   We next turn to investigating the effects of the choice of the regularisation parameter $\varepsilon$ on the quality of the recovered dynamics. To quantitatively measure this, we choose to compute the average correlation $r$ between estimated and ground truth fate probabilities across the three lineages. We applied StationaryOT using both entropy and quadratic regularisations, and let $\varepsilon$ vary on a log-scale from $10^{-2.5} - 10^1$ and $10^{-1} - 10^2$ respectively.

As shown in Fig 5A, in the case of entropy-regularised optimal transport we observe from the fate probability estimates that there is clearly a single optimal value of this parameter at $\varepsilon = 0.026$. This is larger than the theoretically optimal value of $\sigma^2\Delta t = 0.0125$, in keeping with our observations discussed earlier (see Choice of $\varepsilon$ and $\Delta t$.). However, StationaryOT with the theoretically optimal value fares only slightly worse and is located close to the maximum. When $\varepsilon$ is chosen too small or too large, performance degrades. On the other hand, we find that performance when using a quadratic regularisation is much less sensitive to the choice of $\varepsilon$, with the correlation over $\varepsilon$ showing a much flatter profile. We emphasise that $\varepsilon$ is shown on a logarithmic axis in order to remove differences in the scales of $\varepsilon$ for different regularisations.

Since flux rates $R$ are also parameters that need to be specified, we examine the sensitivity to varying the flux rate in Fig 5B. We systematically perturb the proportion of particles that exit at each of the wells $\{z_0, z_1, z_2\}$ by scaling the ground truth flux rates by values in the simplex. We observe that performance is optimal near $(1/3, 1/3, 1/3)$ corresponding to no perturbation to the ground truth flux rates, and degrades moderately as bias is introduced to each well. We show results for entropic and quadratic regularisation where $\varepsilon$ is chosen to be the optimal

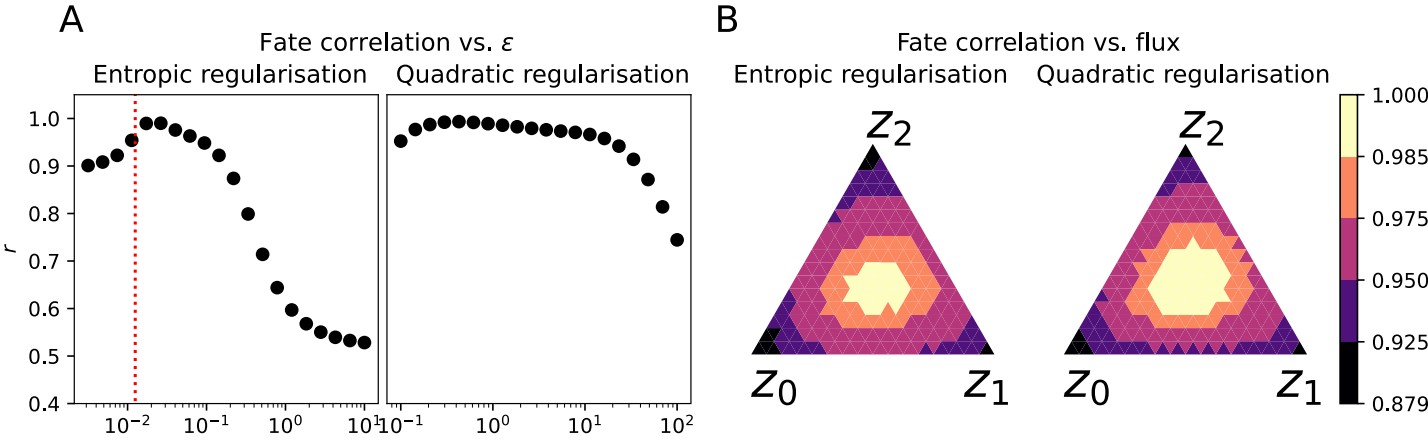

**Fig 5. Effect of parameter choices on inference for potential-driven system.** (a) Correlation for varying regularisation parameter $\varepsilon$ for entropic and quadratic regularisations. For entropic regularisation, the theoretically optimal value of $\varepsilon$ is indicated in red. (b) Summarised correlations for systematic perturbation of flux rates towards each of the wells $\{z_0, z_1, z_2\}$. Note that the simplex represents the perturbation applied to the true flux rates rather than the flux itself, so the centre $(1/3, 1/3, 1/3)$ of each simplex corresponds to using the true flux rates.

values of 0.026 and 0.43 respectively, and note that both choices of regularisation behave very similarly. While it is certain that perfect knowledge of the flux rates yields optimal performance, examining the level sets at $r > 0.95, 0.975$ leads us to conclude that StationaryOT still provides informative fate probabilities for a wide range of perturbations. Interestingly, the correlation remains reasonable even at the corners of the simplex, when all the flux is localised at a single well. This is due to particles in the vicinity of the other wells that diffuse into those wells randomly (as opposed to due to an inferred drift).

**Laws on paths.** The SDE described by Eq (1) naturally induces a probability measure on the space of continuous functions valued in $\mathcal{X}$, from which one can sample cell trajectories. We discuss this point of view at length in related work on the non-stationary case [23]. From this perspective, we may treat the recovered process as inducing a law on discrete-time paths valued in $\bar{\mathcal{X}}$, and we expect that a good estimate of the dynamics should correspond to a law on paths that is closer to the ground truth law. To illustrate this, in Fig 6A we display 100 sample paths over $T = 25$ timesteps, i.e. $t \in \{0, \Delta t, 2\Delta t, \ldots, (T-1)\Delta t\}$. The ground truth paths are obtained by sampling solutions to the Eq (1) using the Euler-Maruyama method with the initial condition Eq (15). To sample paths from the output of StationaryOT, we sampled first $X_0$ to be a random source cell and then let $X_{k\Delta t}, 1 \le k \le 24$ evolve following the Markov chain defined by the transition matrix $P$ output by StationaryOT.

We compare the ground truth to the StationaryOT output for both entropic and quadratic OT for optimal and sub-optimal (taken as $10\times$ the optimal value) choices of the regularisation parameter $\varepsilon$. Visually, it is clear that StationaryOT using both entropic and quadratic OT produces very similar output resembling the ground truth when $\varepsilon$ is chosen to be optimal. On the

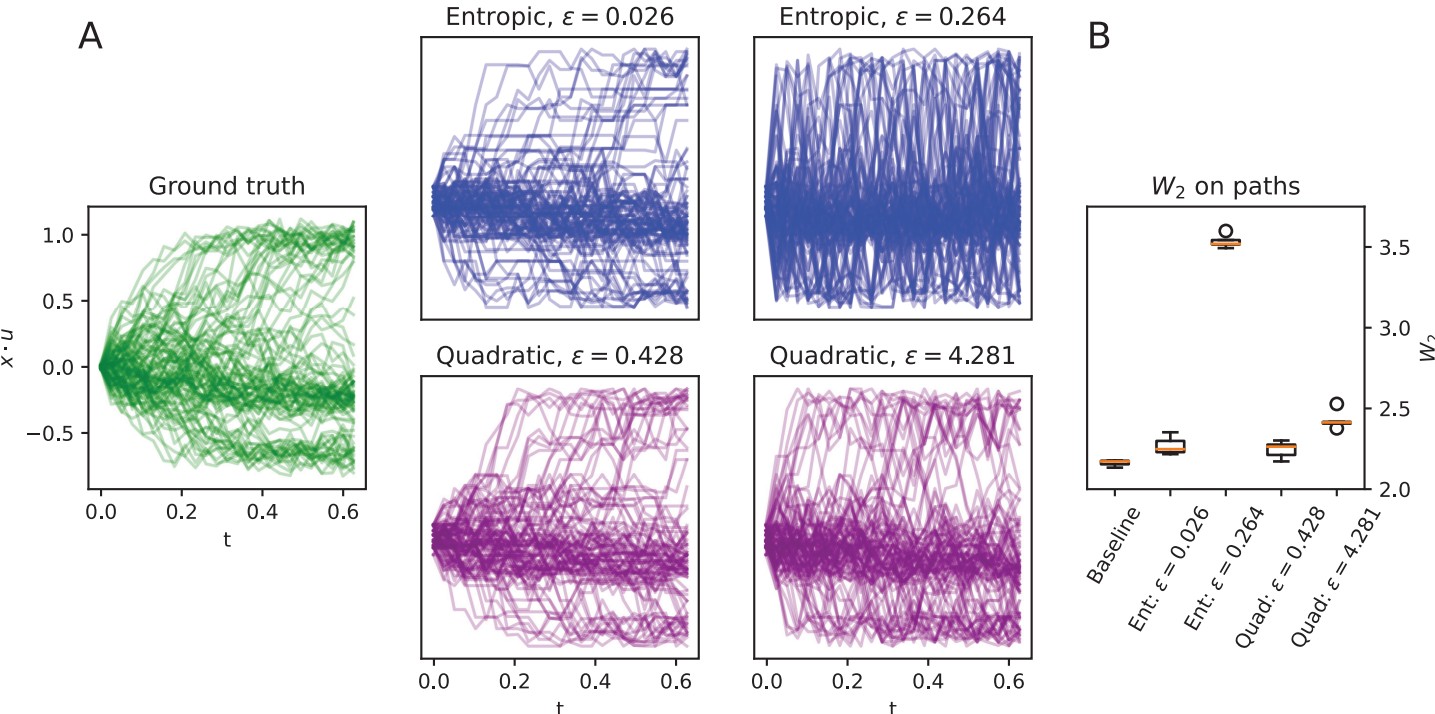

**Fig 6. Inferred dynamics in the space of paths for potential-driven system.** (a) Collections of 100 sample paths from the ground truth process Eq (1) as well as StationaryOT outputs for both entropic and quadratic OT with optimal and sub-optimal $\varepsilon$. The vertical axis corresponds to a projection $\langle x, u \rangle$ of the 10-dimensional state space $\mathcal{X}$ onto a convenient 1-dimensional subspace defined by $u = (\cos(\pi/12), \sin(\pi/12), 0, \ldots, 0)$. (b) $W_2$ error on paths for StationaryOT reconstructions, shown for 5 repeated samplings of 250 paths.

other hand, when $\varepsilon$ is chosen to be too large we observe a visible worsening of performance, with more paths jumping between branches. As we also observed in terms of fate probabilities, the performance of StationaryOT with quadratic regularisation appears to degrade more gracefully than with the entropic regularisation.

To provide a quantitative assessment of performance, the natural metric to use is the 2-Wasserstein ($W_2$) distance on the space of laws on paths, as we also argue in [23]. We refer the reader to S1 Appendix for details on the definition of the $W_2$ distance. Since we work in the setting of discrete time steps, the squared Euclidean ($L_2$) distance between a pair of paths $f$, $g$ is taken to be

$$\|f - g\|_2^2 = \frac{1}{T}\sum_{k=0}^{T-1}\|f(k\Delta t) - g(k\Delta t)\|_2^2. \tag{17}$$

Using the $W_2$ metric for laws on paths, we computed the error of each reconstruction relative to the ground truth. Importantly, we note that since we are dealing with finite samples, the expected $W_2$ distance between independent collections of sample paths from the same distribution will be nonzero. Thus, as done in [23] we compute a baseline error as the $W_2$ distance between independent samplings of 250 paths from the ground truth. In Fig 6B we show the average $W_2$ error over 5 resamplings of 250 paths, from which we note that StationaryOT with entropic or quadratic OT yields results that are close to the baseline in $W_2$ error when $\varepsilon$ was chosen to be optimal. On the other hand, picking $\varepsilon$ to be too large leads to a higher error for both methods, but with entropic OT performing significantly worse than quadratic OT.

## Simulated data—Non-conservative dynamics

Now we consider the case where the drift $v(x)$ is no longer the gradient of a potential landscape, i.e. there is a curl component. In this case, the underlying process is no longer identifiable from only sampled spatial locations [18, 23], and it is necessary to have additional velocity estimates in order to estimate cellular trajectories.

**Simulation setup and parameters.**   To illustrate this, we consider a process with a drift field given by the sum of a potential-driven term and a non-conservative vector field, i.e.

$$v(x) = -\nabla\Psi(x) + f(x). \tag{18}$$

Again, we work in $\mathcal{X} = \mathbb{R}^{10}$ and we take

$$\Psi(x) = \exp\left(-\frac{x_1^2 + x_2^2}{h^2}\right) + \frac{1}{2}(x_1^2 + x_2^2) + 10\sum_{i=3}^{10}x_i^2 \tag{19}$$

$$f(x_1, x_2) = 10\exp\left(-\frac{x_1^2 + x_2^2}{h^2}\right)\begin{bmatrix} \cos(\theta) & -\sin(\theta) \\ \sin(\theta) & \cos(\theta) \end{bmatrix}\begin{bmatrix} x_1 \\ x_2 \end{bmatrix}. \tag{20}$$

We pick $h = 0.5$, controlling how rapidly the field $f$ decays and the location of the potential well in $\Psi$. In the first two dimensions of $\mathcal{X}$, particles can be thought of as diffusing on a radially symmetric potential field with a ring of wells located about the origin, and subject to a superimposed anticlockwise vector field that decays away from the origin. We show a surface plot of $\Psi(x)$ and a vector field plot of $f(x)$ in Fig 7A.

We initialise particles following the initial distribution $X_0 \sim 0.01\mathcal{N}(0, 1)$ that are then subject to the drift-diffusion process with diffusivity $\sigma^2 = 0.1$. The minimum of the circular potential well is located along a cylinder of radius 0.721 about the origin in the first two dimensions

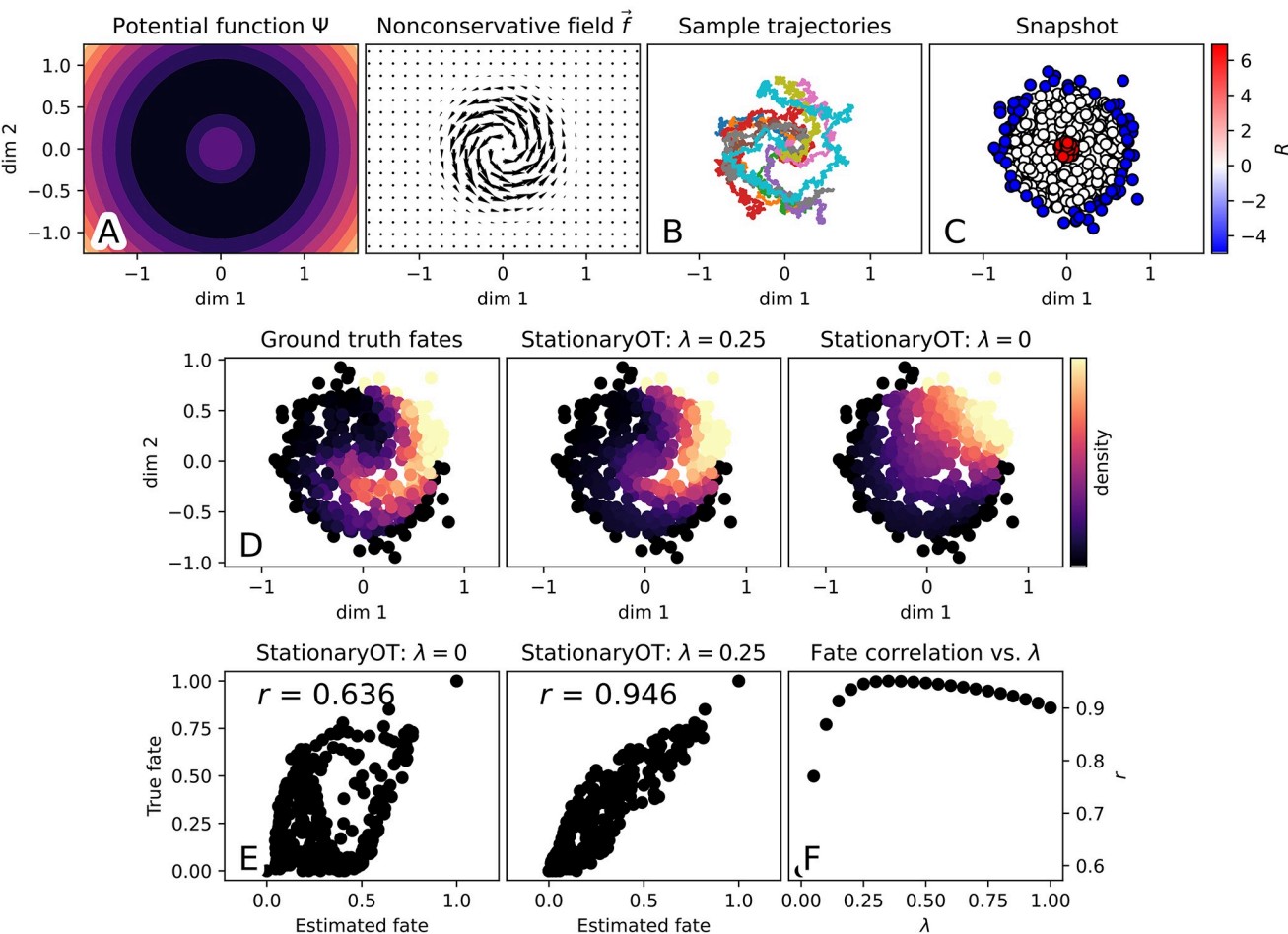

**Fig 7. Non-conservative simulation.** (a) Illustration of potential-driven ($\Psi$) and non-conservative ($f$) components of the overall drift $v$. (b) Examples of simulated particle trajectories $X_t^{(i)}$ following the drift-diffusion process. (c) Snapshot samples shown in the first two dimensions of $\mathcal{X}$, with source ($R > 0$) and sink ($R < 0$) regions indicated. (d) Comparison of fate probabilities towards the sinks in the first quadrant. (e). Correlation of estimated fate probabilities to ground truth fates with ($\lambda = 0.25$) and without incorporation of velocity data ($\lambda = 0$). (f) Summary of fate probability correlation as a function of $\lambda \in [0, 1]$.

of $\mathcal{X}$, and we treat all points outside this cylinder as a sink region, in which particles are removed at exponential rate 5. We sample 500 particles from this process and designate cells found within a ball of radius 0.1 about the origin to be source cells, and cells located in the sink region to be sink cells. Sink cells were assigned a flux rate $R_i = -5$, and source cells were assigned a uniform flux rate so that $\Sigma_i R_i = 0$, as in the previous example. We illustrate in Fig 7B some example trajectories from this simulation, and in Fig 7C we display the sampled snapshot $\hat{\rho}_{eq}$ along with the flux rates.

**StationaryOT with and without velocity data.** For each sampled cell $x_i$, we obtain velocity estimates by evaluating the drift vector field $v(x_i)$ at its location. We then formed two cost matrices: $C_{euc}$, the matrix of squared Euclidean distances, and $C_{velo}$ the matrix of cosine similarities as defined in Eq (13). Both matrices were normalised to have unit mean. Note here that this normalisation is purely an empirical choice, and no corresponding normalisation of the cost was performed in the potential-driven case because of the theoretical motivation in the potential-driven case.

We constructed the optimal transport cost matrix to be a convex combination of the Euclidean and velocity cost matrices:

$$C = (1 - \lambda)C_{\text{euc}} + \lambda C_{\text{velo}},$$

and we took $\lambda = 0.25, 0$, respectively corresponding to StationaryOT with and without velocity information. Both entropic and quadratic OT were used to solve for couplings, with $\varepsilon = 0.05$ and $\varepsilon = 0.5$ respectively. Since the setting of this simulation is rotationally invariant in the first two dimensions, we choose to summarise our results in terms of the absorption probabilities for cells entering the region

$$\mathcal{S} = \{x_i : \Theta(x_i) \in [0, \pi/2] \text{ and } R_i < 0\},$$

i.e. the set of sink cells in the first quadrant in $(x_1, x_2)$. As shown in Fig 7D, the ground truth fate probabilities clearly capture the rotational component of drift, with the set of cells fated towards $\mathcal{S}$ forming a curled shape. We observe that StationaryOT with velocity data produces results qualitatively capturing this effect, whilst neglecting velocity information leads to a symmetric fate profile that reflects only the potential-driven component as expected. To quantitatively compare fates, we computed as previously the Pearson correlation between the estimated fate probabilities and the ground truth. We show this in Fig 7E, from which we observe that StationaryOT with velocity data produces a markedly improved fate correlation ($r = 0.953$) compared to StationaryOT without velocity data ($r = 0.631$). Finally, in Fig 7F we show the fate correlation $r$ as a function of the parameter $\lambda \in [0, 1]$ that controls the composition of the cost matrix $C$. The correlation improves rapidly as $\lambda$ is increased from 0 and attains a maximum before it declines slowly as $\lambda$ is further increased towards 1. We conclude that even relatively small choices of $\lambda$ can greatly improve the accuracy of the inferred fate probabilities.

**Laws on paths.** As in the case of the potential-driven system, we may examine sample paths from the ground truth process as well as the estimates output by StationaryOT. We sample trajectories with the initial condition

$$\pi_0 = \{x_i : \Theta(x_i) \in (-\pi/6, \pi/6) \text{ and } \|x_i\| \in (0.25, 0.5)\}.$$

We illustrate these in Fig 8 in the first two dimensions of $\mathcal{X}$. Again, we observe that incorporation of velocity estimates yields results that clearly reflect the rotational trajectories in the ground truth. On the other hand, without using velocity information, we observe sample paths consistent with only the potential-driven component. Additionally, for either choice of regularisation we observe that StationaryOT overestimates the rotational drift as cells settle into the potential well. This effect can be attributed to the fact that the cosine similarity cost of Eq (13) depends only on the orientation of the rotational field, and thus is unaware of its decay as cells drift towards the well. In this situation, we can only expect to capture the rotational field qualitatively rather than quantitatively. We suggest that possible remedies for this effect may include weighting entries of $C_{\text{velo}}$ by velocity magnitudes or using an alternative velocity cost that is based on squared Euclidean distances.

**Sensitivity to noise.** Finally, are interested in investigating the behaviour of StationaryOT when the provided velocity estimates are subject to additive noise, that is

$$\hat{\boldsymbol{v}}(x_i) = \boldsymbol{v}_0(x_i) + \alpha\eta\mathcal{N}(0, I)$$

where $\alpha$ is a scale constant chosen such that $\boldsymbol{v}_0/\alpha$ has order 1, i.e. the noise term is on the same order as the signal. We pick $\alpha = \mathbb{E}_{x_i}\|\boldsymbol{v}_0(x_i)\|$. We applied StationaryOT using both entropic

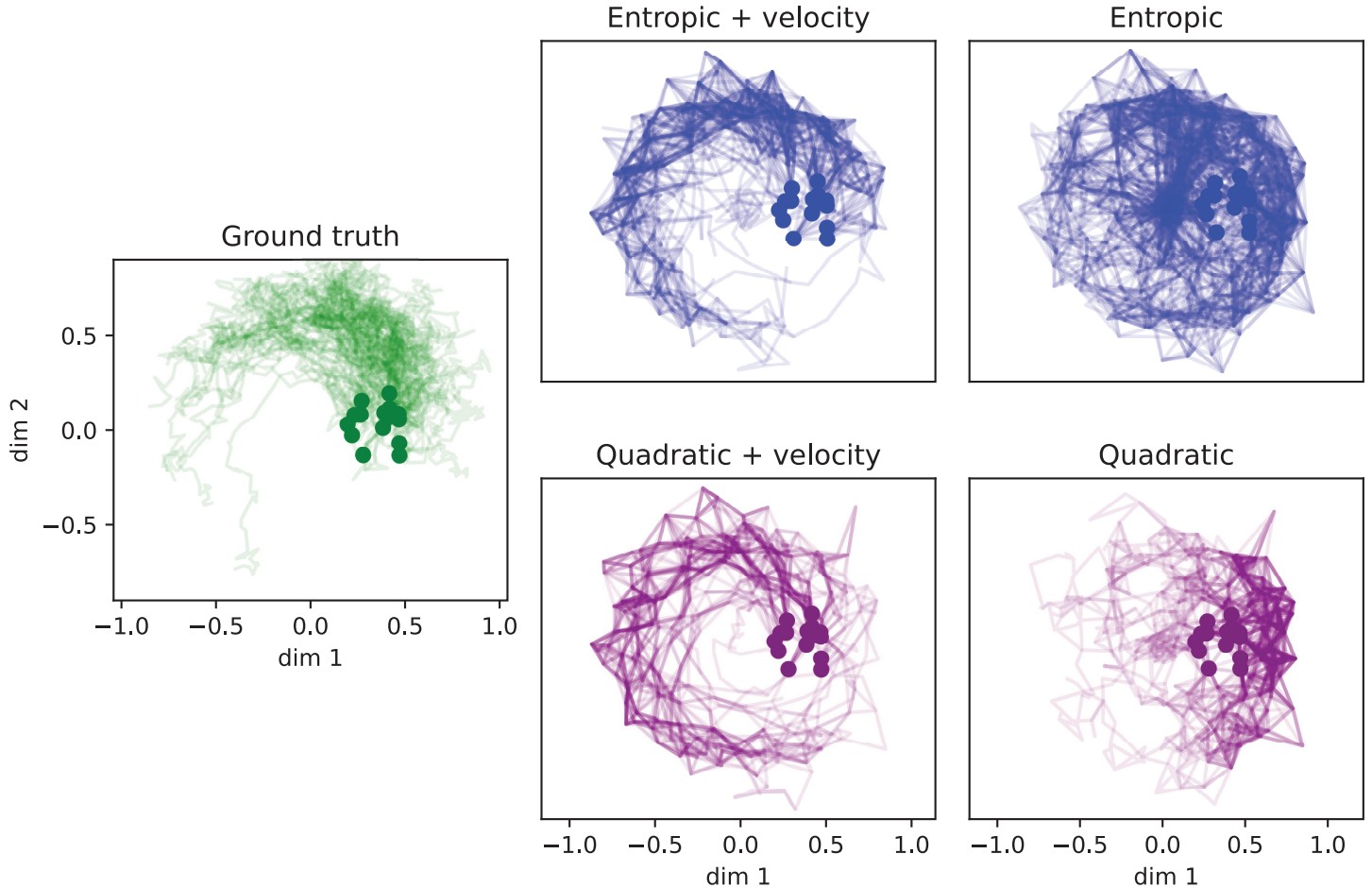

**Fig 8. Inferred dynamics in the space of paths for non-conservative system.** Collections of 100 sample paths drawn from the ground truth process Eq (1), as well as StationaryOT output with and without velocity estimates for both entropic and quadratic OT. We indicate the initial condition $\pi_0$ as dots.

and quadratic OT for values of $\eta \in [0, 2]$ and choices of regularisation $\varepsilon$ chosen in the range $10^{-2} - 10^0$ (logarithmic) for entropic OT and $0.5 - 10$ (linear) for quadratic OT.

For additional comparison, for each noise level $\eta$ we also computed a transition matrix based solely on cosine similarities of velocity estimates to $k$-nearest neighbour ($k$-NN) graph edges using the scVelo package [20] in which the transition law for each cell $x_i$ is

$$\mathbb{P}[X_{\Delta t} = x_j | X_0 = x_i] \propto \exp\left(\kappa \cos \angle (x_j - x_i, v_i)\right), \quad x_j \in \text{neighbours}(x_i). \quad (21)$$

In the above, $\kappa$ is a scale parameter controlling the level of directedness in the resulting transition law, with larger $\kappa$ corresponding to increased directedness in the transition law. We used $\kappa$ in the range 2.5–25 and all other parameters were taken to be defaults.

In each case, performance was summarised as we did previously in terms of the fate correlation for the set $\mathcal{S}$. We show results summarised over 10 independent repeats in Fig 9A and we observe that, as expected, performance degrades for all methods as the level of noise increases. However, StationaryOT with either entropic or quadratic regularisation consistently produces more accurate fate estimates compared to the scVelo method. We argue that this effect reflects the fact that StationaryOT is a global method and solves for transition laws that best agree with the inputs across the dataset. On the other hand, the scheme described by Eq (21) is local in

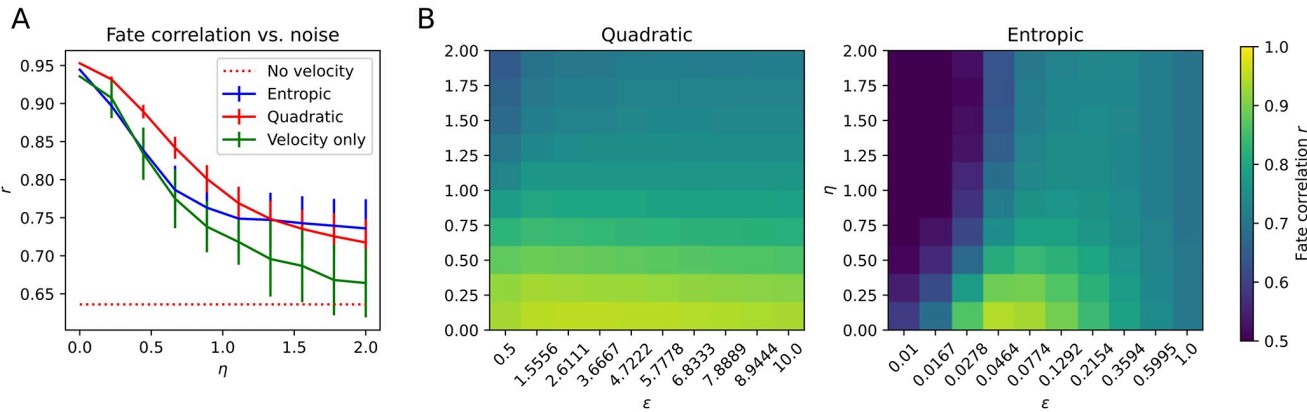

**Fig 9. Effect of parameter choices on inference for non-conservative system.** (a) Correlation of estimated fate probabilities to ground truth as a function of noise $\eta$. (b) Sensitivity of entropic and quadratic regularisations to the choice of $\varepsilon$.

that the transition law for a single cell can be determined by only considering a single velocity vector and a few neighbouring locations.

Additionally, for moderate levels of noise ($\eta \in [0, 1]$), we observe that the quadratic regularisation outperforms the entropic regularisation. Roughly speaking, we suspect that this results from the fact that the transition laws recovered by quadratic OT are concentrated on a sparse subset of cells, limiting the effect that an error in any single velocity estimate can have on the overall Markov chain. On the other hand, since entropic regularisation yields dense transition laws, errors can be propagated across the full support of $\bar{\mathcal{X}}$.

In Fig 9B we examine the sensitivity of StationaryOT performance on the choice of the regularisation $\varepsilon$. As we observed in the potential-driven setting, the entropic regularisation appears to depend strongly on the choice of $\varepsilon$ whereas a quadratic regularisation behaves similarly across the values of $\varepsilon$ used.

### *Arabidopsis thaliana* root tip scRNA-seq data

**Overview.** We now apply StationaryOT to the scRNA-seq atlas dataset generated by Shahan et al. [42], which comprises of gene expression data from $1.1 \times 10^5$ cells from the first 0.5 cm of the *Arabidopsis thaliana* root tip. Stem cells occur close to the tip of the root and differentiate into ten distinct lineages (see Fig 10), with cells becoming increasingly differentiated as they increase in distance from the stem cells. Additionally, the terminal 0.5 cm of the root captures all tissue developmental zones, including the root cap, meristem, elongation zone, and part of the maturation zone. While new cells are constantly produced in the meristem, the bottom 0.5 cm is expected to be in equilibrium as cell division and elongation push existing differentiated cells out of the 0.5 cm section of interest, preserving a constant profile of cell types as illustrated in Fig 10A. Lineages and developmental zones are shown anatomically on the bottom 0.5 cm of the root in Fig 10B as well as on a UMAP embedding of the dataset in Fig 11.

**Application of StationaryOT.** For each cell $x_i$, daily growth rates $g_i$ were estimated from imaging data of the growing meristem over a week-long period [43]. Using these growth rates and the proportion of cells expected to be actively dividing, we estimated that roughly 5% of the cells in each lineage would be replaced in a 6-hour period ($\Delta t = 0.25$) and selected the 5% of most differentiated cells from each lineage as sinks, as defined by pseudotime. For these sink cells we set $g_i = 0$, i.e. they are completely removed over the time interval. The remaining

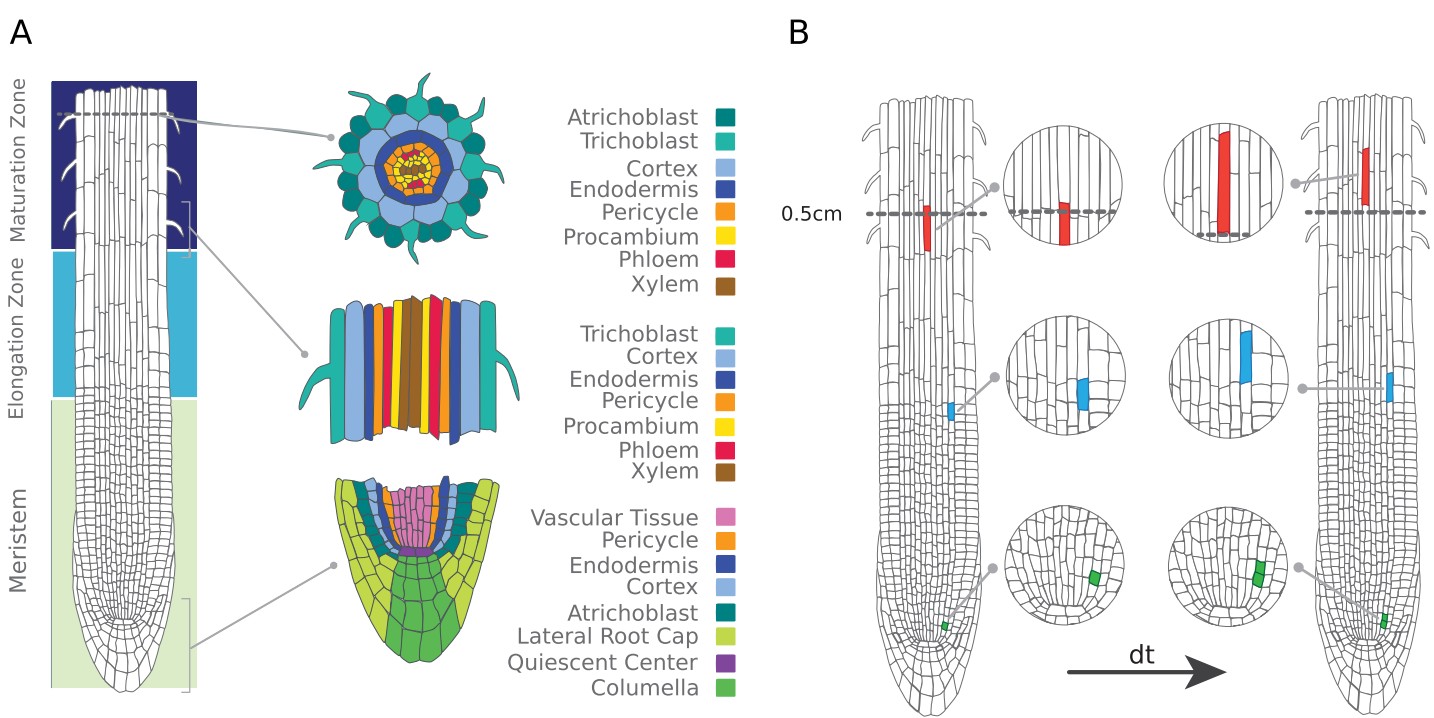

**Fig 10. The *Arabidopsis* root tip system.** (a) While individual cells divide (green), elongate (blue), and are displaced from the bottom 0.5 cm (red) as the root grows, cell populations remain in equilibrium. (b) The structure of the *Arabidopsis thaliana* root tip by developmental zone (left) and lineage (right) (Illustrations modified from the Plant Illustration repository [44]).

cells were assigned a flux rate $R_i$ chosen to agree with the biological growth estimates, i.e. for each cell $x_i$ we take $R_i$ such that $\exp(R_i) = g_i$.

We applied StationaryOT using both entropic and quadratic regularisations with parameters $\varepsilon = 0.025\bar{C}$ and $\varepsilon = 2.5\bar{C}$ respectively, where the scale factor $\bar{C}$ is taken as the mean value of the squared Euclidean cost matrix $C$. We found our results to be robust to changes of a

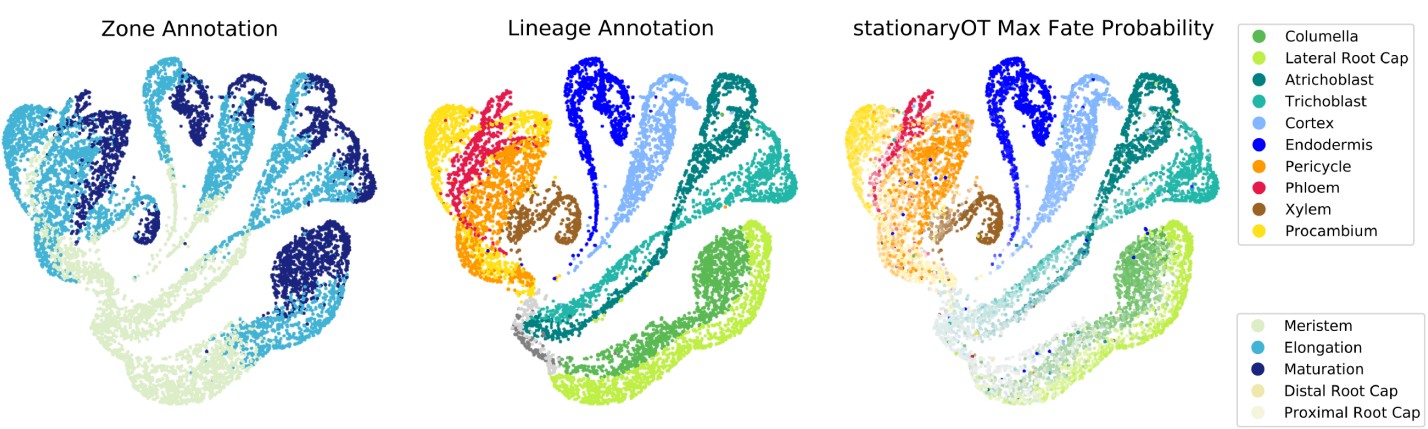

**Fig 11. *Arabidopsis* atlas cell annotations and StationaryOT output.** Developmental zone (left) and lineage annotations (centre) shown on a UMAP embedding. Putative fate probabilities from StationaryOT with entropic regularisation are visualised on the right, where each cell is coloured by putative fate and its saturation based on the magnitude of that probability. For over 80% of cells the putative fate matched the annotation, with the magnitude of the probability increasing later in development.

factor of two in the number of sinks, the time step size, and $\varepsilon$. Due to computational limitations of the standard implementation of the method, we applied StationaryOT to a subset of 10,000 cells sampled from the full dataset, though we also demonstrate two methods to scale the analysis to the full atlas. For completeness, we display in UMAP coordinates fate probabilities for each lineage in S3 Fig.

In *Arabidopsis* root development, cell lineage is fixed early in development [43]. Thus, for each cell $x_i$ we may regard the lineage $j$ corresponding to the largest fate probability, i.e. arg-max$_j$ $p_{ij}$ as its putative fate. We checked whether these putative fates matched the manually curated atlas annotation, and used the magnitude of the corresponding fate probability, $0 \leq p_{ij} \leq 1$, as a measure of the confidence of prediction. StationaryOT with quadratic and entropic regularisation performed similarly in terms of the percentage of cells where the putative fate matched the atlas annotation, matching 81% and 80% of cells respectively (see Fig 12). Both regularisations also performed similarly in terms of the magnitudes of the putative fates, with the entropic regularisation achieving an average of 69%, increasing from an average of 40% for cells in the meristem to an average of 84% for cells in the maturation zone and quadratic achieving an average of 65%, ranging from 38% in the meristem to 80% in the maturation zone (see S2 Fig). This trend can also be seen clearly by visualising the maximum fate probability of each cell on the UMAP (see Fig 13). As in many cases external estimates of growth are not available, we created alternate growth rates based on cell cycle genes. Despite differences between the estimates, StationaryOT performed similarly matching 79% and 78% of cells for entropic and quadratic regularisation respectively (see S1 Appendix).

Both choices of regularisation performed well on nine of the ten lineages, struggling only with mature procambium cells, as shown in Fig 12. We believe this is due to an inconsistency between the pseudotime and developmental zone annotations, where cells in the elongation zone received higher pseudotimes than cells in the maturation zone, resulting in them being incorrectly set as terminal states (see S1 Fig). Both regularisations were robust to changes in parameters, where the percentage of cells whose putative fate matched the annotation changed

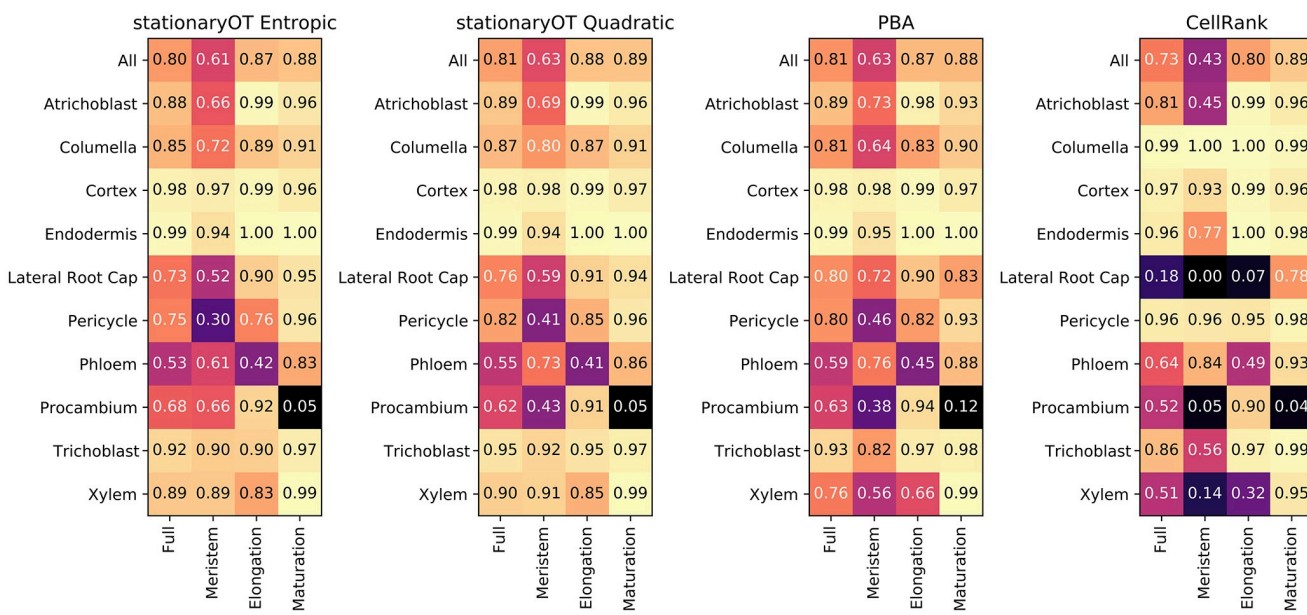

**Fig 12. Comparison of StationaryOT performance to other methods.** Proportion of cells where the maximum probability matches the annotation by developmental zone and lineage.

by no more than 2% when changing a parameter by a factor of two. StationaryOT with quadratic regularisation was particularly robust, with performance degrading by no more than 2% when multiple parameters were changed by up to a factor of five (see S1 Appendix).

**Comparison to PBA.** Since population balance analysis (PBA) [18] addresses the same problem as StationaryOT, it is natural to evaluate its performance on the *Arabidopsis* root dataset. We show results for 1% sinks, $\Delta t = 0.25$ (6-hour time step), diffusivity $D = 2.5$, and $k = 10$ for the $k$-NN graph. These parameters were found to yield the best results over a parameter sweep (see S1 Appendix). PBA was on par with the StationaryOT methods, with 81% of putative fates matching the annotation, compared to 80% and 81% for the StationaryOT analyses (see Fig 12). Average fate probabilities were also similar, with PBA achieving an average of 64% compared to 65% and 69% for the StationaryOT methods (see S2 Fig). Like StationaryOT, the average fate probabilities increased as the tissue matured (see Fig 13). Given that PBA and StationaryOT are methodologically distinct, the fact that they perform similarly is a strong indication that the results reflect the underlying biology, rather than artefacts from the respective models.

In general however, we found PBA to be more sensitive to parameter values than StationaryOT. Assigning 5% of cells in each lineage as sinks for a 6-hour time step ($\Delta t = 0.25$) is

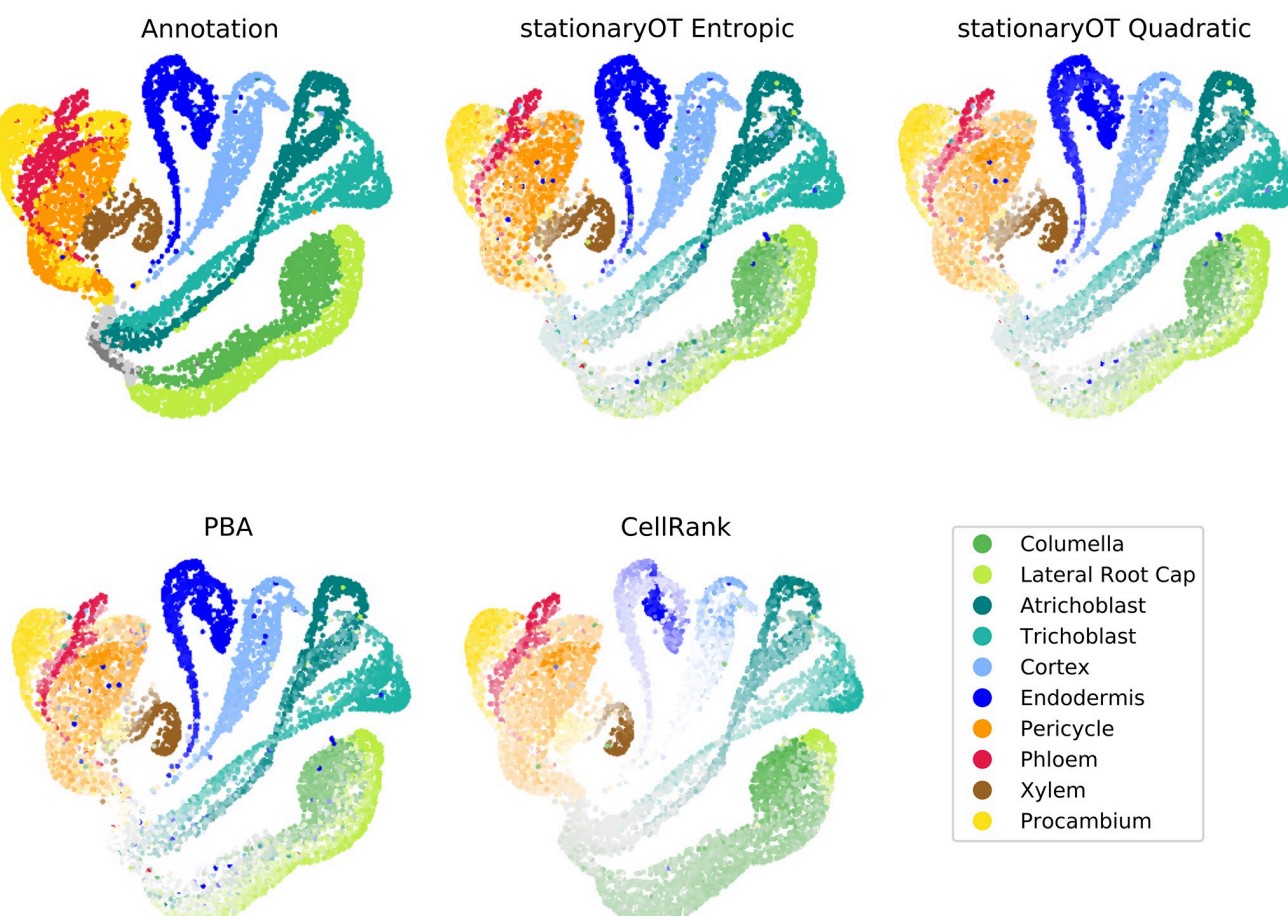

**Fig 13. Comparison of fate probabilities found by StationaryOT to other methods.** Fate probabilities for StationaryOT with entropic and quadratic regularisations compared to the annotation, as well as PBA and CellRank output. The colour indicates the maximum fate probability (putative fate) of each cell and the colour saturation shows the magnitude of the fate probability.

biologically motivated in order to balance the number of cells created due to growth with the number of sinks. Using this sink selection scheme, PBA was found to perform poorly for the columella lineage, incorrectly assigning all columella cells a putative fate of lateral root cap. This lowered the overall percentage of putative fates that matched the annotation to 72% (see S1 Appendix). Only through an extensive parameter sweep did we find the combination of parameters that resulted in columella cells receiving the correct putative fate. We found PBA to be generally more sensitive to other parameter changes, matching 7% fewer cells to the annotation when a single parameter was changed by a factor of two compared to only 2% for the StationaryOT methods (see S1 Appendix). This may be of concern, since in many applications there may not be sufficient prior biological knowledge to distinguish between good and bad parameter choices.

**Comparison to CellRank.**    CellRank is a trajectory inference method that uses both transcriptomic similarity and RNA velocity data to estimate transition laws for cells [22]. The method consists of three key steps: computing cell state transition probabilities, inferring macrostates from the resulting Markov chain, and computing fate probabilities to these macrostates. For ease of comparison between other methods discussed here, our summary will focus mostly on the computation of transition probabilities.

CellRank computes a transition matrix from a $k$-NN graph using a combination of transcriptomic similarity and RNA velocity data. First, a $k$-NN graph is computed using cell transcriptomic similarities and then symmetrised. Edge weights are assigned based on similarity estimates between neighbouring cell states. The resulting graph is then converted into a matrix containing similarity estimates between neighbours. For each cell, transition probabilities are calculated from RNA velocity data by considering the correlation of the RNA velocity vector with displacement vectors corresponding to edges in the $k$-NN graph. These correlations are used to create a categorical distribution on the neighbours of the cell, giving transition probabilities. To better account for noise in the velocity data, the final transition probabilities are taken to be a linear combination of velocity-based probabilities and similarity-based probabilities.

We applied CellRank to the same 10,000 cell subset of the *Arabidopsis* dataset used for the StationaryOT and PBA analyses. Using the output transition matrix, we computed fate probabilities and assigned putative fates as previously described. In terms of putative fates, we found that CellRank matched 73% of cells to the atlas annotation, compared to 80% and 81% achieved by the StationaryOT analyses. The main differences occurred in the lateral root cap and xylem tissues (see Fig 12). CellRank also had less confidence in fate prediction, having an average fate probability of 45% compared to greater than 60% for all other methods (see S2 Fig) and probabilities remained low through the elongation zone (see Fig 13). Finally, we note that CellRank uses a mixture of a directed transition matrix derived from RNA velocity and an undirected transition matrix computed from expression similarity.

## Computing fates for large datasets

The running time for StationaryOT depends on the number of cells, with the main computational costs (in the case of the entropic regularisation) arising from (1) Sinkhorn iterations involving a series of matrix-vector products, and (2) solving a system of linear equations to compute fate probabilities. Computational cost therefore scales roughly quadratically in the number of cells (at least for a fixed number of iterations) and we found that datasets of up to $10^4$ cells could be processed directly using a straightforward implementation of the method. For completeness, we show in S5 Fig the computation time as a function of number of input cells, on a standard CPU-only Google Colaboratory instance (Intel Xeon, 2.30 GHz x2, 12 GB

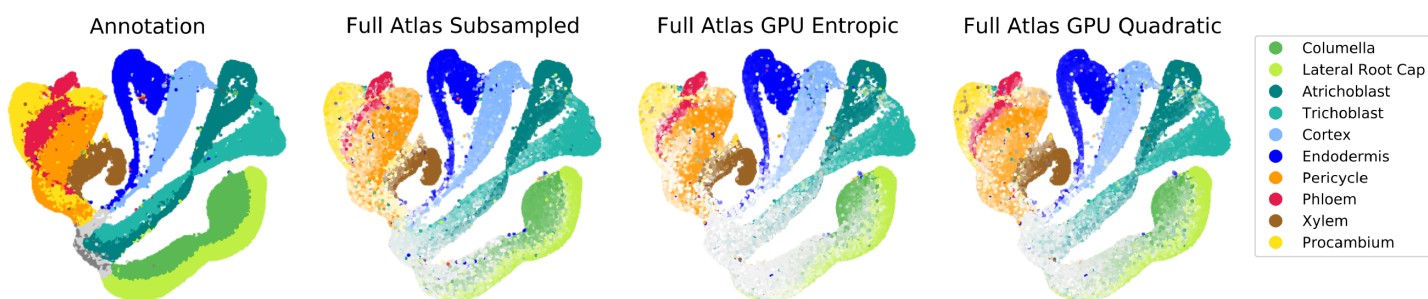

**Fig 14. Output of StationaryOT on full atlas dataset.** Atlas annotation on the full dataset ($1.1 \times 10^5$ cells) shown in UMAP coordinates compared to fate probabilities computed on the full dataset respectively using the subsampling approach (using entropic regularisation for each subproblem) and memory-efficient GPU implementations of StationaryOT with entropic and quadratic regularisations.

RAM). In order to compute cell fates for datasets with very large numbers of cells we propose two approaches.

**Repeated subsampling.** We first randomly partition the dataset of interest into $k$ subsets of size approximately $10^4$, or such that the computation time for StationaryOT is acceptable. Fates are computed for each subset, and this procedure is repeated $j$ times with repeated random partitioning. We then average the computed fates on a cell-by-cell basis, to produce aggregated fate probabilities.

We applied this approach to the full $1.1 \times 10^5$ cell atlas, partitioning it into 10 subsets and applying StationaryOT separately to each subset (see Fig 14). This was repeated 10 times to account for sampling error. Between the fates found directly for each subset and the consensus fates in the full atlas, 97% of cells shared the same putative fate and the maximum fate values had a correlation of 0.96. Accounting for all fate values, the correlation rose to 0.99.

**Memory-efficient GPU implementation with KeOps.** For both entropic and quadratic regularisations, algorithms for solution of the optimal transport minimisation problem can be implemented using the KeOps library [45] so as to avoid storing all $N \times N$ matrices in memory. Along with GPU acceleration, this allows StationaryOT to be applied directly to datasets with many more cells than can be handled by the standard implementation due to memory constraints.

In the case of entropy-regularised optimal transport, the Sinkhorn algorithm can be implemented in a straightforward manner so that the transport plan (a matrix of size $N \times N$) is parameterised in closed form by two dual variables $u$ and $v$ (vectors each of length $N$) [31]. From here, one may construct a linear system to solve for fate probabilities of the form $Ax = b$ where $A$ is again parameterised by the dual variables $(u, v)$ and thus not explicitly stored in memory. For quadratically-regularised optimal transport, a similar representation of the problem in terms of dual variables holds. We provide an implementation of the semi-smooth Newton method proposed in [41, Algorithm 2] that utilises the KeOps library. As mentioned earlier, quadratically regularised optimal transport has the property that the transport plans (and hence the transition laws of cells) will be sparse. In addition to being more interpretable, for large numbers of cells this means additional computational advantages, especially in terms of directly storing the entries of the sparse transition matrix, and computing the fate probabilities.

We applied StationaryOT to the full $1.1 \times 10^5$ cell *Arabidopsis* root dataset using the KeOps implementation with GPU acceleration, using both entropic and quadratic regularisations. We found that solution of the optimal transport problem took roughly 15 and 20 minutes using entropic and quadratic regularisations respectively. We used the same parameter choices

as used for the subsampled dataset. For an entropic regularisation, the resulting transportation plan was dense (effectively all entries were nonzero). In contrast, the quadratic regularisation yielded a very sparse transportation plan as expected (0.17% nonzero entries). Computation of the fate probabilities for the entropic regularisation was significantly more time-consuming than for the quadratic regularisation, taking approximately 10 minutes and 2 minutes respectively. We display the fate probabilities for entropic and quadratic regularisations in Fig 14. The difference in the runtimes reflects the fact that, compared to dense systems, sparse systems of linear equations can be solved much more efficiently using iterative methods. We compared the fates found for a 10,000 cell subset to the fates for StationaryOT with both entropic and quadratic regularisation on the full atlas and found them to perform similarly. For both methods, 90% of cells shared the same putative fate as the 10,000 subset and both had a 0.97 correlation for fate magnitudes accounting for all fates. With the entropic regularisation, the putative fate values were slightly higher correlated with the 10,000 cell subset, achieving a correlation of 0.92 compared to 0.87 for quadratic. All computations for the full atlas dataset were done on a Google Colaboratory instance with a 16GB NVIDIA Tesla V100 GPU.

## Discussion

### Summary of our contributions

Optimal transport has been shown to be a widely applicable tool to the problem of trajectory inference in the setting where multiple time points are available [3, 23, 27, 34–36]. We demonstrate that optimal transport can be applied in a natural way to the stationary setting, where a single snapshot of a system at steady state is observed. The framework that we develop is theoretically justified and is naturally motivated by the Waddington's landscape analogy. Furthermore, our scheme boils down to a convex optimisation problem for which there are efficient and well-known methods of solution. The problem can also be generalised to incorporate additional information such as estimates of velocity. Motivated by these observations, we have developed a computational method which we call StationaryOT and show that it can scale to datasets of up to $10^5$ cells.

We demonstrate the efficacy of this method both on both real and simulated data. We find that in practice our method achieves similar performance to that of Population Balance Analysis (PBA) [18], but StationaryOT appears to be less sensitive to parameter choices and is capable of handling additional information such as velocity estimates. Since StationaryOT and PBA are methodologically distinct, the observation that both methods yield similar conclusions is strong evidence that the outputs reflect genuine biological signal, as opposed to artefacts of the methodology.

Overall, we have shown optimal transport to be a common framework for trajectory inference in the setting of both stationary snapshots and non-stationary, time-series data. This provides a unifying perspective for two problems that have traditionally been approached with separate methods.

### Prospects for future work

In terms of future work, there are many potential avenues for extension of the present work. One major direction is the development of generative models, which can extrapolate information about the potential landscape beyond those cell states measured in experiment. We expect that the optimal transport perspective will be important for this, both conceptually and practically. Another relevant problem is that of examining the evolution of systems that are stationary on short timescales but nonstationary on large timescales—for instance, developmental biological systems such as haematopoiesis in humans are stationary on a fast timescale, but

undergo changes on a slow timescale as individuals age. Finally, one could incorporate lineage-tracing to improve trajectory inference, as we have recently done in the non-stationary case [34].

## Supporting information

**S1 Appendix. Theory and methodology supplement.** Supporting theoretical material and root atlas application details.
(PDF)

**S1 Table. Arabidopsis cell cycle signature genes.** Genes used to compute cell cycle signature scores for the Arabidopsis dataset.
(CSV)

**S1 Fig. All four methods poorly matched the annotation for maturation procambium cells (see Fig 12).** We believe this occurred due to a disagreement between the pseudotime and zone annotations, where procambium cells in the elongation zone were given a higher pseudotime than those in the maturation zone, resulting in cells from the elongation zone incorrectly being set as terminal states.
(TIF)

**S2 Fig. Average fate probabilities matching the annotation by cell type and zone for both StationaryOT methods, PBA, and CellRank.**
(TIF)

**S3 Fig. Lineage annotation compared to cell fate probabilities for both StationaryOT methods, PBA, and CellRank.**
(TIF)

**S4 Fig.** Terminal states found using automatic detection functionality offered by the CellRank package, coloured by their corresponding lineage (right). No terminal states were identified for the phloem and procambium lineages. Additionally, as is clear from pseudotime (left), some states that are intermediate are miss-classified as terminal.
(TIF)

**S5 Fig. Computation time for StationaryOT as a function of number of cells for the Arabidopsis dataset on a standard CPU-only Google Colaboratory instance.**
(TIF)

## Acknowledgments

We would like to thank Rachel Shahan, Che-wei Hsu, and the Benfey Lab for their help in understanding the biological context of the *Arabidopsis* root. G.S. would like to thank Allon Klein for an inspiring discussion.

## Author Contributions

**Conceptualization:** Stephen Zhang, Tetsuya Matsumoto, Geoffrey Schiebinger.

**Formal analysis:** Stephen Zhang, Anton Afanassiev, Laura Greenstreet, Tetsuya Matsumoto, Geoffrey Schiebinger.

**Funding acquisition:** Geoffrey Schiebinger.

**Investigation:** Stephen Zhang, Anton Afanassiev, Laura Greenstreet, Tetsuya Matsumoto, Geoffrey Schiebinger.

**Methodology:** Stephen Zhang, Anton Afanassiev, Laura Greenstreet, Tetsuya Matsumoto, Geoffrey Schiebinger.

**Software:** Stephen Zhang, Anton Afanassiev, Laura Greenstreet.

**Supervision:** Geoffrey Schiebinger.

**Validation:** Anton Afanassiev, Laura Greenstreet.

**Visualization:** Anton Afanassiev, Laura Greenstreet.

**Writing – original draft:** Stephen Zhang, Anton Afanassiev, Laura Greenstreet, Geoffrey Schiebinger.

**Writing – review & editing:** Stephen Zhang, Anton Afanassiev, Laura Greenstreet, Geoffrey Schiebinger.

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
