## [Decision Letter · Decision Letter 0]

14 May 2021

Dear Zhang,

Thank you very much for submitting your manuscript "Optimal transport analysis reveals trajectories in steady-state systems" for consideration at PLOS Computational Biology.

As with all papers reviewed by the journal, your manuscript was reviewed by members of the editorial board and by several independent reviewers. In light of the reviews (below this email), we would like to invite the resubmission of a significantly-revised version that takes into account the reviewers' comments. We cannot make any decision about publication until we have seen the revised manuscript and your response to the reviewers' comments. Your revised manuscript is also likely to be sent to reviewers for further evaluation.

Sincerely,

Andreas Spiegler

Guest Editor

PLOS Computational Biology

Douglas Lauffenburger

Deputy Editor

PLOS Computational Biology

Reviewer's Responses to Questions

**Comments to the Authors:**

Reviewer #1: Inferring differentiation dynamics at the single cell level from single-cell measurements is an important and active area of research. Among many existing approaches, optimal transport has been previously shown, by one of the authors, to provide good estimates for differentiation dynamics from single-cell measurements profiled at different time points. They now demonstrate that optimal transport can also be when there is only a snapshot of the single cell measurements, provided that the systems is stationary and a rough estimate of the growth and death rate is available at different regimes of the manifold. They show that this inference is well-justified theoretically when the differentiation dynamics is driven by a potential landscape, while also providing a heuristic extension to non-equilibrium setting. In the latter case, the inference is only qualitatively correct. The authors benchmark their method with existing methods, like PBA, and show comparable performance. However, they provide a quite distinct perspective/approach, and this development will be highly interesting to the community of optimal transport, and also the community of single-cell method development. Overall, this paper is very well written. I therefore highly recommend the publication of this paper after the authors fix the following issues.

Major questions

* I think it is quite a clever idea to split the dynamic equation (2) into the growth equation and also the drift-equation. This makes the application of OT becomes natural. Intuitively, I think it makes a lot of sense, but I just wonder if there is a simple proof that the split is equivalent to the original equation? I think adding this proof will make the paper more solid theoretically.

* Although it makes sense that the inference would work exactly only for a dynamic system driven by a potential field, I could not see this limitation directly. If you start from a population equation that is equivalent to (2), but also allows rotation (non-potential term), you can still perform the splitting as in Fig. 1. And the argument that it should be diffusion-driven at small interval is still valid. Then, you can still apply optimal transport as defined in (8). Maybe I missed it. Could you clarify this in the main text?

* For people to use the package, it would be useful if you could provide a plot showing the computational cost in a standard lab top as a function of the cell number.

* Related to Fig 10, please provide a plot systematically showing the effect of lambda on the inference.

* Also, did you try StationaryOT to the hematopoiesis data that PBA was originally applied? This is just a suggestion. You can decide whether to do this or not, although I think this application will make the package more appealing, and more people would want to try it out.

Minor issues

* In many places the color bar are missing. Please fix that.

* Similarly, the x-label, and y-label are often too difficult to understand. It would be useful to provide a verbal explanation before the annotation. For example, in Fig. 8a, I do not understand what is X*U.

* In the section ‘’Laws on paths”, I do not understand how the trajectory is generated for each single celll (Fig. 8), and also what is the W2 distance for? Is it a distance between path probability distribution from different methods (i.e., inferred by Stationary-OT and ground-truth)? I think part of the problem is that many notations are introduced without explanation. For example, what is f, g and T in the equation above Fig. 8. Please fix this.

Reviewer #2: The authors present a new method for computing trajectories in a steady state systems with a single population snapshot using optimal transport. This method uses the fact that at the steady state equilibrium the population drift plus the growth rate is in equilibrium, i.e. equal at all points. They use this to infer the population drift, represented as a Markov process over the cells, from prior knowledge on the growth rates or source and sink states of cells. This is effectively the inverse of the problem tackled in velocyto and scVelo, which infer sources and sinks from velocity / transition information. Overall, I found this to be a well founded and valuable contribution to the trajectory inference literature in from scRNA-seq data.

The authors claim: “We show that optimal transport analysis, a technique originally designed for analysing time-courses, may also be applied to infer cellular trajectories from a single snapshot of a population in equilibrium. ”

However, a key component of their method requires prior estimates of the growth rate / sink probability of each cell. In fact all of the experiments are performed in the case where we have access to accurate estimates of growth rates, which is not a trivial piece of information to extract from scRNA-seq data. For the root tip scRNA-seq experiment the authors use growth rates estimated from imaging, which is a relatively reliable piece of side information that is not (in general) present in most scRNA-seq experiments. To be more widely applicable it would be important to demonstrate that this method works with unreliable estimates of the growth rate, preferably attainable from Transcriptome data only. Therefore, either the main claim should be modified to inferring trajectories from a snapshot with prior knowledge of per cell growth rates, or additional experiments should be performed to validate the method when growth estimates are more unreliable and realistic.

For the sensitivity comparison of entropic vs. quadratic regularized OT it would be good to mention the relative size of H(\\gamma) and \\| \\gamma \\|_2^2, as if they are of significantly different magnitude (on these examples) then this would have a confounding effect on the sensitivity to epsilon. Otherwise this is quite an interesting observation.

A few notes on related work, it would be useful to draw distinctions between the methods benchmarked in [1] and the trajectory terminology used here. As there are so many different methods for inferring trajectories, it is important to understand where this particular method is most applicable. [2,3] also learn the potential landscape based on approximating the SDE, but do so from time series in continuous (but relatively low dimensional) space. Here the authors approximate the SDE over the empirical distribution. It would be helpful to include some discussion on the costs / benefits of this discrete approximation. Furthermore, [2] discusses estimation of the drift from a single observation of the steady state equation and the drawbacks of this type of inference. It would be useful to add this type of discussion in light of more recent trajectory inference methods. In light of these related works, to my knowledge this method is novel, particularly in building a Markov process from a single snapshot based on growth rates over the support of the empirical distribution.

Minor Comments:

1.2: “Typically we will take X = R^d”, is this true? From my understanding this paper takes X as the support of the data, \\bar{X}

On line 146 you switch between \\mu_0, \\mu_1 notation and \\mu, \\nu notation.

There is an error in the equation above line 205, z_1 is repeated twice, and the z_2 should probably be cos(3 \\pi / 2) as depicted in Figure 3.

What are the units of the estimated MFPT in Figure 6(a), Markov process steps? Or is it normalized by \\delta t?

Line 471 mentions 15-20 minutes for computation on 10,000 cells, it would be nice to mention the hardware used here as this computation is quite hardware sensitive. Thank you for mentioning the hardware used for the full atlas.

Given that you identified sparsity as a key benefit of quadratic regularization, did you try the unregularized OT formulation? This would obviously not scale particularly well, but may be of interest on small examples if sparsity in the transportation matrix is beneficial.

1. Saelens, W., Cannoodt, R., Todorov, H. & Saeys, Y. A comparison of single-cell trajectory inference methods. Nature Biotechnology 37, 547–554 (2019).

2. Hashimoto, T. B., Gifford, D. K. & Jaakkola, T. S. Learning Population-Level Diffusions with Generative Recurrent Networks. in Proceedings of the 33rd International Conference on Machine Learning 2417–2426 (2016).

3. Sisan, D. R., Halter, M., Hubbard, J. B. & Plant, A. L. Predicting rates of cell state change caused by stochastic fluctuations using a data-driven landscape model. Proceedings of the National Academy of Sciences 109, 19262–19267 (2012).

Reviewer #3: In this paper Zhang et al. introduce the use of optimal transport (OT) for trajectory inference in tissues where the differentiating cells are in equilibrium, noting its success in inferring trajectories of cells in synchronous development systems such as embryogenesis. They frame the problem in the context of differential equations at the cell- and population level. The latter measures the change of population densities of cells at a given point in gene expression space as a function of time. An equilibrium state implies setting this population balance PDE equal to zero, which is achieved by sources and sinks in the vector field, that can be interpreted as stem cells dividing to generate new developing cells (sources), and cells either dying or leaving the tissue after maturation (sinks). Optimal transport is used to find a coupling between two cells after a small time step, accounting for these sources and sinks.

While there is a great abundance of trajectory inference tools, the authors have motivated the novelty and added value of their method well. Its ability to incorporate RNA velocity is also a huge plus.

This paper would be a valuable addition to PLOS Computational Biology, but some comments must be addressed first:

Major

I see many similarities with dynamo (https://github.com/aristoteleo/dynamo-release) but it is not referenced in the manuscript. This warrants a comparison, at least in the introduction. Also, can they be integrated by using the vector field recovered by dynamo, which can exploits RNA metabolic labeling and protein acceleration?

A welcome addition to the paper would be an additional section in the introduction to explain the relevant concepts at a higher level. The jump from section 1.1 to 1.2 is currently too large for many readers to appreciate the methodology. Terms such as “potential landscape” and “vector field” should be explained here, as right now they are suddenly mentioned in section 1.1 (line 54) without any explanation. The new section should ideally also explain how OT (section 2.2) is related to the differential equations (1.2). Example papers that use OT in single cell technology with digestible explanations of the methods are novoSpaRc (Nitzan et al., 2019) and SpaOTsc (Cang and Nie, 2020). See also the first results section / Box 1 in the dynamo preprint (Qiu et al., 2021) for a high-level explanation on what is a vector field and what can be done with it.

Minor

Figure 1 needs more explanation in the caption. Please walk the reader through the figure as the idea of matching the two marginal distributions is what motivates the use of OT.

Solution p_G above figure 1: what is g(x)? I don't see it defined and its not referenced anymore.

L134: "the dynamics are Markov". Is my understanding correct that this implies that the cell state at timepoint t+delta t is fully determined by the cell state at time t? I.e. we do not need previous states (markov property). Is it therefore assumed that a cells next state can be predicted fully from its current state, or is this well known? Please elaborate this Markov part and if my understanding is correct, state the assumption more clearly or source the claim.

With RNA velocity we can approximate X_t+delta t. We can use this to capture the transitions of the cells. As you explain in the paper, the main issue is that we do not have the cells state at time t+delta t. So to what extent does this simplify the methodology? Related to this, it is not immediately obvious how RNA velocity can be used, nor is it actually applied in the Arabidopsis thaliana root tip experiment as far as I can tell. Please clarify.

Line 48: "To date, the methods developed for these two different paradigms have remained largely distinct, with time-course methods generally sharing little in common with methods designed for stationary systems. In this paper we show that optimal transport analysis, a technique originally designed for analyzing time-courses [29], may also be applied to infer cellular trajectories from a single snapshot of a population in equilibrium. Therefore, optimal transport (OT) provides a unified approach to inferring trajectories, applicable to both stationary and non-stationary systems."

Clarify what makes a method uniquely applicable for each problem type. The ability to add prior information? I am familiar with GrandPrix (Ahmed et al., 2019) and Ouija (Campbell and Yau, 2019) where a prior can be set on the pseudotime to be roughly the same as capture time.

I appreciate how the growth rates and sink locations are well-motivated in the Arabidopsis thaliana root tip example by referring to the imaging study. However I suspect in many cases this information might not be available. Can you supply fallback mechanisms for such cases? For example, the sinks might be identified using the diffusion end points as done in the original RNA velocity paper (La Manno et al., 2018, fig 3c).

In the root tip example, all non-sink cells are set to be sources, which if my understanding is correct implies that these cells are actively dividing and therefore introduce new particles into the system. From the original paper I understand that only the cells within the meristem divide: “Young, dividing meristematic cells are at the base of each branch followed by elongating and finally mature, differentiated cells at the tips(Fig 1c)”. Please clarify, or, if this decision was a practical one, users would appreciate this as a practical tip in the github repository.

**Have the authors made all data and (if applicable) computational code underlying the findings in their manuscript fully available?**

Reviewer #1: Yes

Reviewer #2: Yes

Reviewer #3: Yes

PLOS authors have the option to publish the peer review history of their article (what does this mean?). If published, this will include your full peer review and any attached files.

Reviewer #1: **Yes: **Shou-Wen Wang

Reviewer #2: No

Reviewer #3: No
---

## [Decision Letter · Decision Letter 1]

20 Sep 2021

Dear Zhang,

We are pleased to inform you that your manuscript 'Optimal transport analysis reveals trajectories in steady-state systems' has been provisionally accepted for publication in PLOS Computational Biology.

Best regards,

Andreas Spiegler

Guest Editor

PLOS Computational Biology

Douglas Lauffenburger

Deputy Editor

PLOS Computational Biology

Reviewer's Responses to Questions

**Comments to the Authors:**

Reviewer #1: The authors have answered all my questions. I therefore recommend the publication of this paper!

Reviewer #2: The authors have revised the manuscript to our satisfaction. We have no further comments.

Reviewer #3: The authors have addressed the comments in a satisfactory manner. The introduction has improved in clarity, both in explaining the methodology and in placing the method in a larger context. The growth rate estimation using the cell cycle scores is a useful addition to the paper as it addresses the majority of cases where no experimental data is available. I also appreciate the helpful clarifications on certain topics due to my own lack of knowledge in the area. Overall the revision has tightened up an already strong manuscript. Accept.

**Have the authors made all data and (if applicable) computational code underlying the findings in their manuscript fully available?**

Reviewer #1: None

Reviewer #2: Yes

Reviewer #3: Yes

PLOS authors have the option to publish the peer review history of their article (what does this mean?). If published, this will include your full peer review and any attached files.

Reviewer #1: **Yes: **Shou-Wen Wang

Reviewer #2: No

Reviewer #3: No

---

## [Editor Report · Acceptance letter]

24 Nov 2021

PCOMPBIOL-D-21-00451R1 

Optimal transport analysis reveals trajectories in steady-state systems

Dear Dr Zhang,

I am pleased to inform you that your manuscript has been formally accepted for publication in PLOS Computational Biology. Your manuscript is now with our production department and you will be notified of the publication date in due course.

With kind regards,

Andrea Szabo
